# Distributed right-lateral strain at the northern boundary of the Quito-Latacunga microblock

Nicolas Harrichhausen<sup>1, 2</sup>, Léo Marconato<sup>2</sup>, Laurence Audin<sup>2</sup>, Pierre Lacan<sup>3, 4</sup>, Stéphane Baize<sup>5</sup>, Hervé Jomard<sup>5</sup>, Alexandra Alvarado<sup>6</sup>, James Hollingsworth<sup>2</sup>, Pierre-Henri Blard<sup>7</sup>, Patricia Ann Mothes<sup>5</sup>, Frédérique Rolandoné<sup>8</sup>, and Iván Dario Ortiz Martin<sup>9</sup>

**Correspondence:** Nicolas Harrichhausen (njharrichhausen@alaska.edu)

# Abstract.

Remote sensing and field data suggest distributed right-lateral faulting at the northern edge of the Quito-Latacunga microblock in northern Ecuador and southern Colombia. Off the west coast of Ecuador and Colombia, oblique subduction of the Nazca Plate beneath the South America plate induces northeastward motion of the Northern Andean Sliver relative to stable South America. Recent geodetic studies show this sliver comprises several independent microblocks, with strain accommodated at each of their boundaries. The Quito-Latacunga microblock, located in the densely populated Interandean valley, shows approximately 3 mm/yr of right-lateral strain at its northern boundary. However, which structures accommodate this deformation is unclear. Using available digital terrain models (DTMs), local DTMs derived from Pleiades satellite stereo-imagery, InSAR, Google Earth imagery, and field surveys, we demonstrate deformation at the northern boundary is distributed across several northeast-striking right-lateral faults in Ecuador and Colombia. InSAR shows that a recent 2022 M 5.7 earthquake resulted in line-of-sight displacement of 5 cm to 13 cm along one of the east-northeast striking, right-lateral faults. Offset sediments and glacial features indicate recent earthquakes on two other faults (the Reservoir and Polylepis faults) north of and subparallel with this rupture. Displaced glacial landforms along the Reservoir fault show slip rates between 0.8 and 6.1 mm/yr, suggesting geologic slip rates that could be higher than geodetic ones. Exposures of the Reservoir fault also show evidence for at least three surface rupturing earthquakes with magnitudes between M 6.3 and M 7.0. Inflation at the nearby Chile-Cerro Negro volcano may influence earthquakes on these faults, enhancing slip and earthquake rates. The Polylepis, Reservoir, and July 25th earthquake faults all overlap with the proposed area for the August 15, 1868, M 6.4-6.8 El Angel earthquake, indicating they could be associated with this damaging event.

<sup>&</sup>lt;sup>1</sup>Department of Geological Sciences, University of Alaska Anchorage, USA

<sup>&</sup>lt;sup>2</sup>Univ. Grenoble Alpes, Univ. Savoie Mont Blanc, CNRS, IRD, Univ. Gustave Eiffel, ISTerre, Grenoble, France

<sup>&</sup>lt;sup>3</sup>Instituto de Geociencias, Universidad Nacional Autónoma de México, Blvd. Juriquilla, 3001, 76230 Juriquilla, Querétaro, México

<sup>&</sup>lt;sup>4</sup>Géosciences Montpellier, University of Montpellier - CNRS, Montpellier, France

<sup>&</sup>lt;sup>5</sup>BERSSIN, Institut de Radioprotection et de Sûreté Nucléaire, Fontenay-aux-Roses, France

<sup>&</sup>lt;sup>6</sup>Instituto Geofisico, Escuela Politécnica Nacional, Quito, Ecuador

<sup>&</sup>lt;sup>7</sup>CRPG, CNRS, Université de Lorraine, 54500, Vandoeuvre-lès-Nancy, France

<sup>&</sup>lt;sup>8</sup>Sorbonne Univ., CNRS-INSU, Institut des Sciences de la Terre Paris, Paris, France

<sup>&</sup>lt;sup>9</sup>Servicio Geológico Colombiano, Bogotá, Colombia

40

#### 1 Introduction

Geodetic block models help show where crustal strain is concentrated, and thus have been used to delineate fault systems and help determine seismic hazard in regions where active faults are difficult to map (e.g., McCaffrey et al., 2007; Allmendinger et al., 2009; Benford et al., 2012; Evans, 2022). The high strain regions between geodetically defined elastic blocks, however, vary in width and either can be composed of discrete structures or distributed deformation over a wider area (e.g., Loveless and Meade, 2011; Elliott and Freymueller, 2020). These different boundary types can be considered in earthquake source models for seismic hazard assessment as area sources if fault locations are very uncertain, as single through-going faults capable of hosting very large earthquakes, or as many smaller structures, which may limit rupture size. Given the spacing of Global Navigation Satellite System (GNSS) stations in many parts of the world are often >>10 km (e.g., Mothes et al., 2013; Geirsson et al., 2017), the block models alone cannot predict whether deformation it is focused or more distributed, and where it is focused onto faults. Thus, while highly useful in targeting areas where crustal deformation is occurring, other methods are required to further characterize shear zones.

A geodetic block model of north-western South America has shown that the Northern Andean forearc Sliver (NAS), which moves NE relative to stable South America, actually comprises several independently moving tectonic blocks (Fig. 1, Jarrin et al., 2023). In the northern Andes of Ecuador and southern Colombia, one of these independent blocks, the Quito-Latacunga microblock, is defined on its western and eastern boundaries by mapped fold and thrust belts and oblique shear zones (Eguez et al., 2003; Alvarado et al., 2016; Marinière et al., 2020). However, the structures that accommodate deformation predicted at northern edge of the Quito-Latacunga have not been well constrained. This region, centered on the Ecuador–Colombia border has hosted destructive historical earthquakes (Beauval et al., 2010), and also contains several active volcanoes (Hall et al., 2008). Therefore, mapping active structures is important for hazard assessment and for exploring the interactions between crustal deformation, inherited structures, and arc-volcanism.

On July 25, 2022 a M<sub>w</sub>5.7 right-lateral strike-slip earthquake ruptured a WSW to ENE striking fault close to the northern boundary of the geodetically defined (Jarrin et al., 2023) Quito-Latacunga microblock (IG-EPN, 2022). In this study, we used InSAR to investigate the coseismic surface deformation of this surface rupture and we used field studies to place it in a tectonic context with nearby faults and volcanic centers. By using a combination of remote sensing and field data, we show that the earthquake ruptured one of several parallel right-lateral strike slip faults south of the Chiles-Cerro Negro volcanic complex, which cross-cut glacial moraines. We use cosmogenic dating to estimate the formation ages of the moraines and right-lateral slip rates of one of the faults. Excavations across one of the faults parallel with the recent rupture, reveals at least three Holocene earthquakes offsetting volcanic soils. These faults lie within the proposed epicentral area, and could be responsible for the 1868 M 6.8 earthquake that damaged the city of El Angel. By placing this study in context with parallel right-lateral faults in Colombia and regional deformation revealed by InSAR (Marconato et al., 2024), we show that the northern boundary of the Quito-Latacunga microblock is a wide zone of distributed deformation.

Figure 1. Tectonic setting of the Quito-Latacunga microblock (QL); and GNSS velocities, geodetic block model boundaries, and strain rates from Jarrin et al. (2023). QL boundaries with geologically constrained slip rates are the Quito fault system (Q), the Latacunga fault system (F), the Pallatanga fault (P), the Reventador thrust faults (R), and the Cayambe-Afiladores-Sibundoy fault system (CAS). The inset shows the broader tectonic context of the Northern Andean Sliver (NAS) and its main eastern boundary fault, the Chingual-Cosanga-Pallatanga-Puná fault system (CCPP) and the CAS fault systems. Location of Figure 2 at the northern boundary of the microblock is also shown. Hillshaded DTM in main figure is a Copernicus 30-m DTM from https://dataspace.copernicus.eu/.

# 2 Tectonic Setting

Oblique subduction of the Nazca plate and collision of the Carnegie Ridge off the west coast of northern South America results in 5.8 to 9.5 mm/yr of northeastward motion of the NAS with respect to stable South America (Fig. 1; e.g., Pennington, 1981;

80

Kellogg et al., 1995; Nocquet et al., 2014; Mora-Páez et al., 2019; Jarrin et al., 2023). This sliver motion was originally thought to be mostly taken up by large strike-slip and oblique-slip fault systems along its eastern margin (Pennington, 1981); primarily the Chingual-Cosanga-Pallatanga-Puná fault system (CCPP) in Ecudaor and the Cayambe-Afiladores-Sibundoy (CAS) fault and Eastern Frontal fault (EFF) system in Colombia (Velandia et al., 2005; Tibaldi et al., 2007; Alvarado et al., 2016). However, more recent work suggests that the NAS can be broken down into several smaller, independently moving, microblocks accommodating strain along each of their boundaries (e.g., Audemard et al., 2014; Alvarado et al., 2016; Audemard M et al., 2021; Jarrin et al., 2023). These studies highlight several regions where unmapped active faults accommodating the strain between the microblocks may be capable of hosting large damaging earthquakes.

A region where the Jarrin et al. (2023) geodetic block model suggests substantial deformation is at the northern edge of the Quito-Latacunga microblock near the border between Ecuador and Colombia (Fig. 1 and Fig. 2). The Quito-Latacunga microblock is located at a large (~ 250 km north-south) left-stepping section of the right-lateral eastern boundary of the NAS (Fig. 1, Alvarado et al., 2016). The contractional left-stepping geometry results in  $\sim 2$  to 3.3 mm/yr of east-west compression along the more northerly trending eastern and western boundaries of the microblock and and ~3 to 6 mm/yr right-lateral strain along the more northeasterly trending southern and northern boundaries (e.g., Jarrin et al., 2023). Along the western boundary, the Quito and the Latacunga thrust fault systems have been shown to accommodate much of the predicted compression (Tibaldi and Ferrari, 1992; Lavenu et al., 1995; Fiorini and Tibaldi, 2012; Alvarado et al., 2016; Marinière et al., 2020). On the southeastern boundary, 0.45 - 0.6 mm/yr of active right-lateral slip has been observed along the Pallatanga fault (Winter et al., 1993; Baize et al., 2015, 2020; Harrichhausen et al., 2023) and faults along strike to northeast that transverse the eastern Cordillera of the Andes (Alvarado et al., 2016; Champenois et al., 2017). At the eastern boundary, ∼10 mm/yr of rightlateral slip has been observed along the CAS (a part of the CCPP), and ~4 mm/yr of west–east shortening has been observed on thrust faults near Reventador Volcano (Tibaldi et al., 2007). Along the northern boundary of the microblock, where the ~3 mm/yr of predicted right-lateral strain is accommodated is not clear. Northeast-striking right-lateral Quaternary faults have been documented near the city of Pasto and on the flanks of the Galeras volcano in Colombia (Rovida and Tibaldi, 2005). Geologic and geomorphic studies of two of these structures, the Buesaco and Aranda faults, show 0.7 to 1.6 mm/yr and Aranda 1.1 to 2.6 mm/yr of Quaternary right-lateral reverse slip (Tibaldi and Leon, 2000). However, these faults are >45 km north of the northern boundary of the Quito-Latacunga microblock (Fig. 2, Jarrin et al., 2023). Historic and instrumental seismicity suggest active faulting further south in Ecuador as well, suggesting a wide zone of distributed deformation.

A southward decrease in GNSS velocities across northern boundary of the Quito-Latacunga microblock is consistent with the right-lateral strain predicted in the Jarrin et al. (2023) block model. GNSS stations near Tulcan, and  $\sim 5$  km north of the boundary observe 7.1 mm/yr and 10.6 mm/yr, respectively, of east-northeast motion relative to stable South America, while a station in Pasto, Colombia  $\sim 40$  km to the north indicates 8.4 mm/yr of northeast motion (Fig. 1). Although no stations are located directly south of the boundary, stations  $\sim 25$  km south-southeast of the boundary in Ibarra and Pimampiro show 6.7 mm/yr and 6.3 mm/yr, respectively, of east-northeast motion with respect to stable South America. Thus, given the wide spacing of the GNSS stations the location of the northern boundary of the Quito-Latacunga microblock, and whether strain here is localized on a single structure or is distributed cannot be discerned from the GNSS data alone. Additionally, GNSS

Figure 2. Caption on next page.

90

110

Figure 2. a) Hillshaded Copernicus 30-m DEM showing: the Jarrin et al. (2023) block model boundaries and strain rates; crustal seismicity between 1993 and 2016 (before the 2016, M 7.8 Pedernales earthquake; IG-EPN); the proposed location of the 1868 M6.6 earthquake (Beauval et al., 2010); the locations of the July 25 fault (JF), the Reservoir fault (RF), the Polylepis fault (PF), the Aranda fault (AF), and the Buesaco fault (BF); potentially active faults mapped in this study; and active faults from Alvarado (2012). The approximate locations of the Chiles-Cerro Negro and Galeras seismic swarms are also shown by their names. b) Geologic of the study area showing major bedrock faults and Quaternary-active volcanoes. Geologic compilation from Gómez et al. (2019). Copernicus DEM available at: https://dataspace.copernicus.eu/.

velocities may capture magmatic processes related to recent volcanic activity at the Chiles-Cerro Negro volcanic complex (Ebmeier et al., 2016) and Galeras volcano.

Recent analysis of the interseismic InSAR velocity field in northern Ecuador and southern Colombia shows a better spatial resolution of deformation across the northern boundary of the Quito-Latacunga microblock, and supports a hypothesis that right-lateral deformation is distributed (Fig. 3, Marconato et al., 2024). The eastward velocity field here shows no sharp gradient in velocities across the geodetic block boundary, instead there is a north-to-south gentle reduction in eastward velocities from  $\sim$ 1.3 to  $\sim$ 0.8 mm/yr over  $\sim$ 20 km (Fig. 3b). These estimates however may be influenced by ongoing inflation at Chiles (Ebmeier et al., 2016; Espín Bedón et al., 2025), illustrated by the substantial increase in the eastward velocity field across an axis that extends to the south-southeast from Chiles (Fig. 3a) along with a maximum uplift rate of 2.9 cm/yr southeast of the volcano. Another InSAR analyses by Espín Bedón et al. (2025) show even greater eastward velocities east of Chiles (6 mm/yr), and uplift rates of 15 mm/yr.

Crustal seismicity at the norther boundary of the Quito-Latacunga microblock (1993 to 2016, IG-EPN) also shows diffuse strike-slip and compressive strain. Small (M 2.0) to moderate (M 5.0) earthquakes are dispersed throughout the region and are not clustered along the boundary defined by the Jarrin et al. (2023) block model. Instead, clusters are located near Chiles and Galeras volcanoes, or near the city of Ibarra (Fig. 2). These earthquake swarms have been attributed to volcanic processes at Chiles-Cerro Negro in 2014 (Ebmeier et al., 2016) and Galeras in 1989 (Jiménez et al., 2009), while the swarms near Galeras in 1993 and 1995 have been suggested to have at least a partially tectonic origin (Jiménez et al., 2009). Focal mechanism analyses of this crustal seismicity indicates a south-to-north transition approximately at the Ecuador-Colombia border from compressive to strike-slip faulting with a relatively consistent east-west shortening direction (Arcila and Muñoz Martín, 2020).

Instrumentally recorded and historical earthquakes with M > 5.0, including the recent  $M_w 5.7$  event that is analyzed in this study, have also occurred along the modeled northern boundary of the Quito-Latacunga block. In 1868, a sequence of two large  $M \sim 6.6$  and  $M \sim 7.25$  earthquakes separated by 10 hours occurred in northern Ecuador (Beauval et al., 2010). Damage intensity data for the first  $M \sim 6.6$  earthquake suggests an epicentre in a region between El Angel and Chiles Volcano (Fig. 2a). The subsequent  $M \sim 7.25$  earthquake, which destroyed the city of Ibarra, is thought to have occurred either south or west of Ibarra and along the western boundary of the microblock. In Colombia, eight  $M \sim 5.6$  to 6.3 earthquakes occurred between 1923 amd 1947, causing substantial damage to communities such as Cumbal, Pasto and Túquerres. More recently, during the 2014 earthquake swarm a  $M \sim 5.6$  right-lateral earthquake occurring on a moderately-dipping southwest-striking fault plane

Figure 3. Caption on next page

**Figure 3.** Interseismic deformation over the study area. a) Interseismic InSAR velocity field for East component derived from Sentinel-1 time-series from Marconato et al. (2024) The velocities are computed between 2017 and 2023 and corrected from the long-wavelength post-seismic effect of the 2016 Pedernales megathrust earthquake. White arrows show the GNSS velocity field from Jarrin et al. (2023), computed before 2016. The focal mechanism of the July 25, 2022 earthquake is shown (IG-EPN), however, the deformation shown does not include coseismic deformation associated with the earthquake. b) Swath-profile of InSAR velocities (colored dots), GNSS velocities (white circles) and block model prediction (Jarrin et al., 2023). The extent of the swath-profile is shown in (a) for InSAR and GNSS by solid and dashed rectangles, respectively.

 $\sim 15$  km to 20 km south of Chiles-Cerro Negro. InSAR and Coulomb stress modeling suggest that this earthquake was induced by inflation south of the volcano that could be attributed to volcanic processes (Ebmeier et al., 2016). Finally, on July 25, 2022 a M 5.7 earthquake occurred 9 km north of San Gabriel, causing damage to this and several other nearby cities such as Tulcan and El Angel.

## 120 3 Geologic Setting

The northern boundary of the Quito-Latacunga microblock is located within the Inter-Andean Valley, a large depression filled with Pliocene to Quaternary volcanic and volcanosedimentary rocks, which overlie and obscure a major terrane boundary between oceanic terranes to the west and metamorphic basement rocks to the east (Fig. 2a). The metamorphosed sediments to the east are believed to represent the ancestral western margin of South America (Aspden and Litherland, 1992; Pratt et al., 2005), while the Jurassic to Cretaceous plutons that intrude into these sediments may represent a continental arc (e.g., Aspden and Litherland, 1992; Villagómez and Spikings, 2013; Bustamante et al., 2016; Zapata et al., 2016), or intrusions in an intracontinental rift (e.g., Cediel et al., 2003).

Three oceanic plateaus, which subsquently accreted to the ancient margin from late Cretaceous through to the Paleocene, comprise the basement rocks west of the terrane boundary in the Inter Andean Valley. The San Juan terrane, which accreted to the South American margin at  $\sim 75$  Ma, and the Guaranda Terrane, which then accreted to the new margin outboard the San Juan terrane at  $\sim 68$  Ma, are exposed in our study area (Boland et al., 2000; Hughes and Pilatasig, 2002; Jaillard et al., 2004, 2009). These basement rocks are overlain by siliciclastic sediments likely derived from exhumation during accretion (Jaillard et al., 2004, 2009), and by younger volcanic rocks related to the modern arc (Boland et al., 2000). The two oceanic terranes are bounded by major, steeply-dipping, southwest to northeast-trending suture zones (Fig. 2a, Boland et al., 2000; Hughes and Pilatasig, 2002; Alvarado et al., 2016). These suture zones have been thought to have been reactivated (Guillier et al., 2001) and may accommodate the right-lateral shear and compression resulting from the northeast motion of the NAS. However, Alvarado et al. (2016) have shown that many of these south-southwest to north-northeast trending structures have been abandoned and are now cross-cut by more recent faulting.

The basement lithologies and associated suture zones are overlain by thick deposits of Miocene to Quaternary volcanic and associated volcano-sedimentary rocks deposited by the numerous volcanic centers in the study area (Fig. 2). Several of




these volcanic centers are clustered around the active Chiles-Cerro Negro complex, which hosted minor eruptive activity in the 19th century (Monsalve-Bustamante et al., 2020). In addition, recent activity has also been reported at the Cumbal volcanic complex  $\sim$ 15 km north of Chiles-Cerro Negro. Two other active volcanoes are located to the north in Colombia near the cities of Tuquerres and Pasto ((Fig. 2). The Azufral volcano near Tuquerres has had 6 periods of eruptive activity over the last 220 thousand years, with the most recent being  $\sim$  280 years BP. The Galeras volano is located 9 km west of Pasto, Colombia, and has seen continuous activity since the 16th century (Monsalve-Bustamante et al., 2020). This volcanic activity at the northern edge of the Quito-Latacunga microblock could play a role in the location of deformation and in the onset of earthquakes (e.g., Ebmeier et al., 2016).

Along with recent volcanism, Quaternary glaciations have shaped much of the high-elevation landscape in the Andes of northern Ecuador and southern Colombia. Glaciations in this region occurred several times throughout the Pleistocene (>130 ka BP) with a maximum extent of ice that may have reached down to 2700 m (Smith et al., 2008; Angel et al., 2017). The last glacial maximum occurred in this region 20-12 ka BP, resulting in ice caps and valley glaciers extending down to elevations between 3000 and 3800 m.a.s.l. (Schubert and Clapperton, 1990). Near Quito, ~ 100 km south of the Cerro Negro-Chiles Volcano, several moraines sequences have been identified suggesting glacial advance to below 3700 m.s.a.l. between 20 ka and 10 ka BP (as summarized in Smith et al., 2008; Angel et al., 2017). Additionally, evidence of a an ice cap forming at a mean elevation of 4200 m between 11 and 10 ka BP in the Papallacta valley east of Quito suggest a late stage glacial advance associated with the Younger Dryas (Clapperton et al., 1997). On the plateau surrounding the Chiles-Cerro Negro volcanic complex, U-shaped valleys, cirques, and terminal moraines that are indicative of recent glaciation (e.g., Taussi et al., 2023) are observed at elevations just below 3500 m.a.s.l. Simlarly, Calvache and Duque-Trujillo (2016) recognize glacial landforms on the flanks of the Galeras Volcano in southern Colombia that terminate elevations just below 3400 m.a.s.l. Although no ice caps remain on either of these volcanoes, the landscape suggests substantial glacial influence up to the beginning of the Holocene.

# 4 Methods

To analyze the surface deformation associated with the July 25, 2022 earthquake, we computed coseismic SAR interferograms near El Angel using both Sentinel-1 (C-band) and ALOS-2 ScanSAR (L-band) SAR data. Two Sentinel-1 interferograms, from descending track 142 (2022/07/17-2022/07/29; Fig. 4a) and ascending track 120 (2022/07/15-2022/07/27; Fig. 4b) were produced using the ForM@Ter GDM-SAR-In service (poleterresolide.fr/le-service-gdm-sar-in/) based on the NSBAS processing chain (Doin et al., 2011; Thollard et al., 2021). One ALOS-2 ScanSAR interferogram, from descending track 139 (2022/07/10-2022/08/21; Fig. 4c) was also computed using the GMTSAR software (Sandwell et al., 2011). Note that an ionospheric phase screen obtained using the split-spectrum technique was removed from the ALOS-2 interferogram (Gomba et al., 2015). All interferograms were filtered using a sliding window removing the phase gradient before filtering and reintroducing it afterwards. The ALOS-2 interferogram being very coherent and containing few fringes caused by coseismic displacements, could be unwrapped easily even close to the rupture (Fig. 4d, Doin et al., 2023).





Reconnaissance geomorphic mapping to map potentially active faults was conducted using GoogleEarth Pro (version: 7.3.6.10201), a hillshaded 4-m-resolution lidar- and photogrammetry-derived digital terrain model (DTM) of selected areas in Ecuador from SigTierras of the Ecuadorian Ministry of Agriculture, Quito (http://ide.sigtierras.gob.ec/geoportal/), and a 5-m resolution GeoSAR acquired DTM of southern Colombia from the Geological Survey of Colmbia GeoSAR project (2020) available at https://sgc.gov.co/. Potentially active faults were mapped using QGIS software v. 3.22 (QGIS Association, 2018), by identifying continuous lineaments in the landscape with horizontal or vertical offsets of Quaternary-aged geomorphic features (e.g. glacial moraines and channels). Often, the expected offsets of slowly-slipping faults in young terrain would be expected to be lower than the resolution of the DTMs we used (4-5 m), therefore we also considered lineaments where clear offsets were not visible but there was evidence of topographic benches and associated landslides seen in satellite imagery.

To further assess the potentially active faults we mapped using the lower-resolution DTMs, we conducted detailed geomorphic/geologic mapping using higher resolution Pleiades satellite derived DTMs and field studies. High-resolution (






Samples of 2 cm thickness were collected at the top of flat exposed surfaces of three different andesite boulder, whose coordinates and elevations were precisely noted and are available in the supplemental dataset (Table S1, Harrichhausen et al., 2025). Given their location on the ridge crests, the samples experienced negligible mask shielding. Samples were crushed, sieved and pure pyroxene grains of 100-500 microns were extracted by hand, at the Institut des Sciences de la Terre (ISTerre) at the Université Grenoble Alpes in France.  $^3$ He and  $^4$ He concentrations in the pyroxene grains were then measured at the Centre de Recherches Pétrographiques et Géochimiques (CRPG) in Nancy, France. Pyroxenes were fused in high vacuum at 1500°C using a home designed induction furnace (Zimmermann et al., 2018). After gas purification, helium isotopes abundances were measured with a tuned Split Flight Tube mass spectrometer, following methods described in Blard (2021).  $^4$ He furnace blanks were  $(4.2\pm0.2) \times 10^{-15}$  mol and represented from 3% to 15% of the sample signal, while  $^3$ He blanks were undetectable. Table S1 in the supplemental dataset (Harrichhausen et al., 2025) shows the He isotope abundances. During the same analytical session, we analyzed a CRONUS-P standard material, that yielded an  $^3$ He concentration compatible with the certified value, within analytical uncertainties (Blard et al., 2015).

Given that these volcanic rocks have a Pleistocene age, we assume that both the nucleogenic  ${}^{3}$ He and the radiogenic  ${}^{4}$ He build ups are negligible. We considered that the measured  ${}^{4}$ He is magmatic and used a  ${}^{3}$ He/ ${}^{4}$ He ratio of 5 Ra (Ra =  $1.384\pm10^{-6}$  being the atmospheric ratio) to correct the measured  ${}^{3}$ He concentrations (Blard et al., 2013). Ages were then calculated using the online CREp calculator (Martin et al., 2017) incorporating various  ${}^{3}$ He production rate models and scaling factors. Such approach permits to encompass all the sources of uncertainties associated with this cosmogenic  ${}^{3}$ He exposure dating.

We also located roadcuts that exposed active faults. To examine the earthquake history of this structure, we conducted a detailed analyses of the fault and sediments in the exposures. After cleaning and cutting the exposures to produce a vertical face, we used Agisoft Metashape software (Agisoft, 2021) to create orthophotomosaics from overlapping digitial photos of the exposure (e.g., Reitman et al., 2015). These photomosaics were used to map the exposure on a digital tablet. A very high-resolution photomosaic and trench log of the primary exposure is provided in a dataset (Harrichhausen et al., 2025).

We collected 14 bulk sediment samples for radiocarbon dating to constrain the sediment deposition and earthquake event ages exposed in the roadcuts. Samples were dated at the Artemis AMS-French National facility (CEA Saclay, LMC14; Moreau et al., 2020). We used OxCal version 4.4.4 (Bronk Ramsey, 2021) for radiocarbon calibration with the IntCal20 <sup>14</sup>C production curve (Reimer et al., 2020) and report our results as calendar calibrated ages before 1950 (cal BP) or thousands of calendar calibrated years before 1950 (ka). We then used OxCal to incorporate all available chronological constraints into a model that uses Monte Carlo routines and Bayesian statistics to determine the probability distribution function (PDF) of geologic unit and earthquake event ages (e.g., Lienkaemper and Ramsey, 2009). OxCal codes are available in an archived dataset (Harrichhausen et al., 2025).

#### 5 Results






# 5.1 July 25 Earthquake

On July 25, 2022 a  $M_w$ 5.7 EQ occurred  $\sim$  10 km northeast of El Angel and  $\sim$  15 km southeast of the Chiles and Cerro Negro volcanoes (Fig. 3), near 0.743°N, -77.844°E. This earthquake was shallow, with an hypocentral depth of 

**Figure 4.** Coseismic deformation during the July 25, 2022 earthquake. a,b,c) Sentinel-1 descending, ascending and ALOS-2 descending wrapped interferograms. Line-of-Sight convention is positive when away from satellite. d) ALOS-2 unwrapped interferogram. Line-of-Sight convention is positive when towards satellite. e) Field picture of cracks found close to the surface rupture observed on interferograms, located by the yellow star in (d). f,g,h) Profiles of the Line-of-Sight coseismic displacement from the ALOS-2 unwrapped interferogram. Profiles are located in (d). Blue and red numbers indicates the total amount of deformation and the offset at the surface rupture, respectively.





lineations located north of the city of El Angel, which we have termed the Reservoir and Polylepis faults, show the clearest signs of active deformation (RF and PF in Fig. 2a).

#### **5.3** Reservoir fault zone

Pleiades-derived DTMs show two parallel faults offsetting glacial moraines  $\sim 20$  km west of the epicenter of the July 25 earthquake, and  $\sim 12$  km southwest of Chiles volcano (Fig. 5). Both faults are subvertical, and the northern fault strikes  $\sim 60^{\circ}$  to  $70^{\circ}$  while the southern fault strikes  $\sim 70^{\circ}$ . The surface expression of the northern fault is relatively continuous and crosses the entire width of the Pleiades DTM (10.5 km). It also may continue up to 15 km to the northeast where the low-resolution Copernicus DEM shows northeast-striking lineaments along the southern flank of Chiles. The visible surface trace of the second, southern fault is less continuous, consisting of up to  $\sim 2$  km-long segments that are likely connected at depth. Topographic lineaments visible in the low-resolution Copernicus DEM suggest this southern fault continues to the northeast for up to 10 km from the edge of Fig. 5b. To the southwest, the projection of the more east to west-striking southern fault merges with the northern fault.

Along both fault traces, field reconnaissance and the Pleiades DTMs show up to 3 m-high north- and south-facing scarps and benches, along with sag ponds, and marshes where streams crossed and were dammed by the fault (Fig. 5; Fig. 6). Small (

**Figure 5.** Caption on next page

Figure 5. a) Unannotated hillshaded Pleiades-derived DTM of the Reservoir fault zone northwest of El Angel, and south of Chiles Volcano. b) Annotated version of (a) showing surface traces of the faults visible in the DTM and/or the field (red), right-lateral offsets of glacial lateral moraine ridges (orange), landslides (pink shading), and the largest channels (blue). The road to the reservoir is shown in dark grey, locations of (c) and (d) are shown by green boxes, and the contour intervals are 10 m. c) Slope map the southern strand of the Reservoir fault zone where it cross cuts and offsets a lateral moraine, and has formed a small landslide. Green lines show the different extrapolated moraine lineations, their intersections with the fault strands on either side of the landslide, and the associated offset measurements. d) Slope map of the northern Reservoir fault zone where it is crossed by the road to the reservoir. Three parallel segments (north, middle, and south branches) form the northern fault. Locations of the sag pond and the fault exposures shown in Fig. 6 and Fig. 7 are shown.

Figure 6. a) Photo looking SE at a north, uphill-facing scarp that is  $\sim 2$  m high (see person for approximate scale) on the northern Reservoir fault zone. Location of photo shown in Fig. 5b. b) Interpreted photo (looking south) of roadcut exposure of the southern segment of the northern Reservoir fault (location in Fig. 5d). Geologic legend in Fig. 7. c) Unannoted (left) and interpreted oblique view (right) of the same fault shown in (b). This view shows the vertical fault offsetting Andisol (S1 and S2) and a colluvial wedge (CW2) against massive and poorly bedded, matrix supported diamict (R0 and R1). Bulk radiocarbon sample locations shown with yellow boxes and ages are mean calibrated ages before present (in thousands of years before 1950). See Table 2 for the full dataset of uncalibrated and calibrated radiocarbon ages.

# 5.3.1 Reservoir fault exposures

In the NE corner of Fig. 5b, three segments of the northern fault are crossed by a dirt road accessing a reservoir in the El Angel ecological reserve (Fig. 5d). The roadcuts provide exposure of three branches of the northern fault cross-cutting post-glacial (~ 20 ka) sediments and volcanic soils.

The middle branch of the northern fault, located immediately north of a  $\sim 50$  m wide sag pond, is exposed on the northern side of the road (Fig. 5d; Fig. 7). Three main stratigraphic units are visible here. The oldest unit, R1, is composed of 1 to 10






cm-thick interbedded layers of silt, coarse sand, and minor sub-rounded pebbles within a silt matrix. R2 is a 30 to 50 cm-thick 305 layer of clast-supported sub-rounded to sub-angular pebbles, cobbles, and rare boulders deposited on top of R1. The matrix of this unit is cemented with calcite. Finally, the youngest units, S1-4 are a 0.3 to 1.5 m-thick layer of massive Andisol, organic rich soil formed on Andesitic volcanic ash (e.g., Poulenard et al., 2001).

Each of these stratigraphic units have been deformed by an up to 30 cm-thick, subvertical SE-NW-striking fault zone, which obliquely cross-cuts the roadcut (4.5H, Fig. 7b). This fault is also visible near the bottom of the exposure between 2H and 4H, 310 highlighting its irregular geometry. The fault zone narrows to a single stand above 1.5V and is finally truncated by overlying sediments between 1.8V and 1.9V. The base of the Andisol (S1) is vertically offset by  $\sim$ 1 m to 1.4 m by this fault zone. We observe a sequence of 3 colluvial wedges overlying 10 to 15 cm of Andisol (S1) and Unit R2 in the northern down-thrown block. The oldest colluvial wedge, CW1, is  $\sim 30$  cm thick and is overlain by  $\sim 15$  cm of Andisol (S2) and CW2 is  $\sim 50$  cm thick and overlain by a thin Andisol layer (S3). Both CW1 and CW2 are composed of a mixture of colluvium derived from units R1, R2, and the Andisol (S1-3) and both wedges are cross-cut by the fault. CW2 is offset vertically by  $\sim 30$  cm, indicating there is a third colluvial wedge deposited above it. The location of this colluvial wedge (CW3) is approximate, based on slight discolouration of the Andisol and the presence of large boulders within the Andisol.

In addition to the main fault zone, a SW-NE-striking subvertical fault offets unit S1 vertically, north-side-up, by  $\sim 30$  cm at the northern end of the exposure (10H, Fig. 7b). A small colluvial wedge (CW1) has also formed here and based on the same thickness of Andisol (S1) that underlies this colluvial wedge, it is likely coeval with CW1 on the main fault zone. Finally, in the southern up-thrown block of the main fault zone at 3.75H, a  $\sim$  30-cm wide,  $\sim$  60-cm deep fissure cross-cuts Units R2 and R1 and is filled with Andisol (S1-3).

The fault strand south of the the sag pond is exposed on the southern side of the road to the reservoir (Fig. 5d). At the 2 m-high roadcut (Fig. 6b, c), the basal unit is a massive, poorly-sorted, matrix-supported diamict with sub-angular to subrounded pebbles to boulders (R0). The matrix of this unit is primarily sand and is cemented by calcite. R0 is overlain by a  $\sim 40$  to 60 cm thick layer of silt and sand with minor pebbles (R1), which in turn is overlain by by  $\sim 0.5$  to 1 m of Andisol. A vertical, SW-NE-striking planar fault obliquely crosses the roadcut offsetting units R1 and R0 against the Andisol (S2-3) and a colluvial wedge (CW1). Small tension fractures in the cemented R0 matrix are subvertical and parallel with the road, suggesting right-lateral motion (e.g., Doblas, 1998). Colluvium from R0 and R1, including a large boulder have been deposited on the northern side of the fault (CW1). This colluvial wedge is at least 1 m-thick and is cross-cut by the fault implying a second colluvial wedge overlying the fault. We have approximated the location of the second colluvial wedge (CW2) by the increased cobble abundance in the Andisol at this location, however its thickness and true extent are unknown. Vertical separation across the southern strand of the fault is unknown because the basal contact of the Andisol is not exposed north of the fault. The north-side-up offset exposed at this location is at odds with its location on the southern side of a sag pond, suggesting that the offset exposed here is apparent.

The northern most strand of the northern fault is crossed by the road  $\sim 300$  m east of the middle strand and it is exposed in a  $\sim 3$  m-high roadcut on the southern side of the road (Fig. 5d).  $\sim 0.3$  to 1.0 m-thick layers of massive, cemented diamict (R0 and R2 observed in the other roadcuts) is interbedded with sand and silt (R1) in the bottom 2.5 m of the outcrop. Andisol

Figure 7. Unannotated (a) and interpreted (b) orthophotomosaics of an exposure of the middle segment of the northern Reservoir fault zone. Local grid coordinates are 1 m by 1 m. Outcrop location shown in Fig. 5d. Bulk radiocarbon sample locations shown with yellow boxes and ages are mean calibrated ages before present (in thousands of years before 1950). c) Results of Bayesian model of radio carbon dates showing age ranges for sedimentary units and three earthquakes. The coloured boxes show the maximum possible time range for each stratigraphic unit based on the model. The ages of sample #36 and sample #40 are anomalous as they are out of stratigraphic order likely due to recycling or bioturbation, and the stratigraphic position of sample #48 relative to CW1 is unclear. These sample were not used in the model. Two radiocarbon dates from CW1 are from the exposure of the southern fault branch in Fig. 6c. See Table 2 for the full dataset of uncalibrated and calibrated radiocarbon ages. d) Cartoon illustrating the stratigraphic evolution of the exposed deposits and faults in the exposure shown in (a) and (b).

overlies these units along an erosional disconformity, which also separates the Andisol from the basal units (R0-R2) in the other two roadcuts (Fig. 6b, c; Fig. 7a, b), however, the scale of those outcrops made this observation less obvious. The northern most fault strand is subvertical, strikes SW, and is less than 20 cm thick. There is  $\sim$  20 cm of vertical separation of an R0-R1 contact across the fault, however the fault does not cross-cut the Andisol or offset the disconformity.

### **5.3.2** Dating

360

370

To estimate slip rates and earthquake ages on the Reservoir fault zone we collected 13 bulk sediment samples for radiocarbon dating and two samples from basalt boulders exposed on glacial moraines or landforms for surface exposure dating. The cosmogenic <sup>3</sup>He ages estimate when the offset moraines in Fig. 5 formed, and the radiocarbon ages bracket the ages of the the three earthquakes observed in the roadcuts (Fig. 6 and Fig. 7).

3He ages shows that the exposure of two basalt boulder surfaces located at an elevation of 3925 masl along the northern Reservoir fault, ~ 200 m NE of the roadcut in Fig. 7 and ~ 200 m N of the ridgerest (Fig. 5d), occurred between 11.9 and 20.4 ka (Samples ANG-50 and ANG-51, Table 1). This age range is based on the 95% intervals of ages calculated using three different production rate models for <sup>3</sup>He and scaling models. Slightly younger exposure ages (mean: 14.7 ka) of the basalt boulders are calculated using a high Andes local production rate (Blard et al., 2013; Delunel et al., 2016) compared to ages (mean: 16.4 ka) calculated using a global production rate (Martin et al., 2017). All of these ages compare well with the timing of glacial advance and subsequent retreat in the tropical Andes during Marine Isotope Stage 2 (MIS2), leaving areas below 4000 masl ice free by ~ 15 ka (e.g., Rodbell et al., 2009; Blard et al., 2013; Angel et al., 2017; Martin et al., 2018).

Sediment collected from the exposures of the middle and southern branches of the northern Reservoir fault zone (Fig.6; Fig. 6b) yield mean radiocarbon ages between 2.95 ka and 8.39 ka (Table 2). We use 10 of these ages to build a chronological model of sediment deposition and deformation that brackets the earthquake ages (Fig 7c).

We have removed samples #36, #40, and #48 from the chronological model (Fig 7c) for the following reasons. Sample #36  $(6.51 \pm 0.10 \text{ ka})$ , is located at the very top of unit R1 (Fig. 7). This sample is younger than samples from three units that are stratigraphically above it including: #47  $(8.44 \pm 0.09 \text{ ka})$  from R2, #37  $(7.47 \pm 0.08 \text{ ka})$  from S1, and #38 $(6.60 \pm 0.10 \text{ ka})$  from CW1. Given that the age of #36 is out of stratigraphic order, and is located immediately next to a fissure, it is likely younger organic material introduced through bioturbation. Sample #40  $(6.10 \pm 0.14 \text{ ka})$  from unit CW2, is older than #39  $(5.46 \pm 0.16 \text{ ka})$  taken from the Andisol unit (S1) stratigraphically below it. Because of this age discrepancy and the observation that #39 is part of a colluvial wedge (CW2), we assume it is recycled. Finally, we remove sample #48  $(4.61 \pm 0.19 \text{ ka})$  due to its relatively unknown stratigraphic position relative to the colluvial wedges (Fig. 7b).

Based on the 10 remaining samples, collected from both outcrops of the middle and southern strands of the northern Reservoir fault zone, we estimate the ages of the three earthquakes (EQ1-EQ3, Fig. 7c, d). The first event occurred at  $7.03\pm0.51$  ka, and two subsequent events occurred at  $4.89\pm0.83$  ka and  $4.32\pm0.83$  ka. Because we have no reliable date of the colluvial wedge (CW2) that formed after EQ2, EQ2 and EQ3 have overlapping age distributions. However, given the complete degradation of the scarp that formed CW2 (Fig. 7), we assume a substantial time span occurred from EQ2, during the formation of the upper facies of CW2, to EQ3 (e.g., hundreds of years, Gray et al., 2022).

**Table 1.** <sup>3</sup>He ages from the Reservoir and Polyepis faults computed using the CREp (Martin et al., 2017) calculator.

| Sample  | Scaling Factor | Contributing PR | <sup>3</sup> He exposure Age (ka) | $2\sigma$ (ka) |
|---------|----------------|-----------------|-----------------------------------|----------------|
| Model 1 |                |                 |                                   |                |
|         |                |                 |                                   |                |
| ANG-01  | 7.329060166    | -134.1908617    | 133                               | 21             |
| ANG-50  | 6.453600254    | -134.1908617    | 14.7                              | 2.0            |
| ANG-51  | 6.538672844    | -134.1908617    | 15.44                             | 2.0            |
| Model 2 |                |                 |                                   |                |
|         |                |                 |                                   |                |
| ANG-01  | 7.159050018    | -120.8756495    | 151                               | 40             |
| ANG-50  | 6.59482767     | -120.8756495    | 16.0                              | 3.4            |
| ANG-51  | 6.656923703    | -120.8756495    | 16.8                              | 3.6            |
| Model 3 |                |                 |                                   |                |
|         |                |                 |                                   |                |
| ANG-01  | 8.198087607    | -129.8619798    | 122                               | 13.4           |
| ANG-50  | 7,033230411    | -129.8619798    | 13.9                              | 2.0            |
| ANG-51  | 7.125870867    | -129.8619798    | 14.6                              | 1.8            |

Model 1 uses a local high Andes production rate (PR) (Blard et al., 2013; Delunel et al.,

2016) and the Lal Stone time dependent scaling scheme (Balco et al., 2008).

Model 2 uses a Global PR (Martin et al., 2017) and the Lal Stone scaling scheme (Balco et al., 2008).

Model 3 uses a local high Andes PR (Blard et al., 2013; Delunel et al., 2016) and the LSD scaling scheme (Balco et al., 2008).

Table 2. Uncalibrated and calibrated radiocarbon ages from the Reservoir and Polyepis faults

| Sample | C (mg) | Delta C <sup>13</sup> (%) | pMC      | Err. pMC | Age BP | Err. age BP | from (cal. BP) | to (cal. BP) | μ (cal. BP) | $\sigma$ (cal. BP) | Med. (cal. BP) |
|--------|--------|---------------------------|----------|----------|--------|-------------|----------------|--------------|-------------|--------------------|----------------|
| #02    | 1.48   | -22.70                    | 39.70843 | 0.17248  | 7420   | 35          | 8344           | 8175         | 8253        | 54                 | 8261           |
| #36    | 0.53   | -25.90                    | 49.04934 | 0.18867  | 5720   | 30          | 6627           | 6407         | 6512        | 52                 | 6509           |
| #37    | 1.40   | -22.70                    | 44.24958 | 0.19085  | 6550   | 35          | 7565           | 7362         | 7465        | 42                 | 7462           |
| #38    | 0.75   | -24.00                    | 48.58477 | 0.19010  | 5800   | 30          | 6670           | 6498         | 6599        | 48                 | 6603           |
| #39    | 1.20   | -23.20                    | 55.48448 | 0.19353  | 4730   | 30          | 5580           | 5326         | 5460        | 82                 | 5472           |
| #40    | 0.94   | -24.90                    | 51.56762 | 0.21079  | 5320   | 35          | 6265           | 5996         | 6099        | 68                 | 6093           |
| #41    | 1.70   | -21.60                    | 64.96098 | 0.20877  | 3465   | 30          | 3833           | 3640         | 3741        | 57                 | 3742           |
| #42    | 0.87   | -28.70                    | 75.49367 | 0.24364  | 2260   | 30          | 2344           | 2155         | 2250        | 60                 | 2230           |
| #43    | 0.98   | -26.20                    | 71.11779 | 0.23652  | 2740   | 30          | 2920           | 2762         | 2826        | 37                 | 2822           |
| #44    | 1.41   | -18.40                    | 51.12925 | 0.20811  | 5390   | 35          | 6287           | 6010         | 6193        | 73                 | 6210           |
| #45    | 1.33   | -23.10                    | 49.62574 | 0.18799  | 5630   | 30          | 6486           | 6314         | 6400        | 46                 | 6406           |
| #47    | 0.60   | -23.30                    | 38.60171 | 0.18395  | 7645   | 40          | 8537           | 8378         | 8442        | 44                 | 8431           |
| #48    | 1.58   | -22.20                    | 60.14346 | 0.20823  | 4085   | 30          | 4806           | 4444         | 4607        | 96                 | 4580           |
| #49    | 1.72   | -19.00                    | 70.80843 | 0.26147  | 2775   | 30          | 2952           | 2783         | 2868        | 47                 | 2868           |

pMC refers to percent Modern Carbon. BP refers to years Before Present (1950). Cal refers to radicarbon ages calibrated using OxCal version 4.4.4 Bronk Ramsey (2021) with the IntCal20 <sup>14</sup>C production curve Reimer et al. (2020).

# 5.4 Polylepis fault

A subparallel-striking fault ~3.5 km southeast of the of the fault exposures of the Reservoir fault also shows evidence of a recent rupture. This structure, which we have termed the Polylepis fault, has been previously mapped by Alvarado (2012) as a network of northeast-striking faults that continue for ~17 km (Fig. 2). Satellite imagery show surface deformation indicative of surface rupture along this structure for ~11 km. This evidence includes a visible lineament cutting across glacial moraines, ridges, and hillslopes; fresh landslides along these lineaments; sag ponds forming along the lineament, and small scarps along the lineament observed in the field (Fig. 8).

**Figure 8.** November 11, 2016 satellite images of the Polylepis fault (Google, Image © CNES / Airbus). a) Overview image showing trace of the fault trace, landslides, offset moraines, and locations of oblique views. b) Trace of the Polylepis fault crossing lateral and ground moraine. Fault trace aligns with the southern edge of a small pond and the scarp here may form a dam. c) and d) show clear lineaments east of the offset moraine in (b).

Continuous cloud cover in the region prevented the acquisition of Pleiades satellite imagery suitable for DTM construction, but field excursions allowed analyses of a north-facing scarp cross cutting a lateral moraine and collection of samples from the moraine for dating (Fig. 8a). The scarp is  $\sim 1$  m high at the top of the lateral moraine, and shows no clear lateral displacement. A north-facing scarp is consistent with the small ponds at the valley bottom immediately east of the moraine (Fig. 8b), however,

these ponds may be also have formed from damming by a recessional moraine. <sup>3</sup>He ages shows that the exposure of a large (~5 m diameter) basalt boulder surface located on the lateral moraine ~650 m south of the scarp at an elevation of 3710 masl, occurred between 116 and 171 ka (Sample ANG-01, Table 1). This age is significantly older than the exposure ages we determined at a higher elevation near the Reservoir fault (Table 1), and the LGM between 20 and 12 ka BP (Schubert and Clapperton, 1990). Due to the anomalously old age, we interpret this boulder may have been exposed during or before glacial plucking, transport, and deposition during the last glacial maximum. A radiocarbon sample from the base of the Andisol on the same moraine (Sample #02, Table 2, Fig. 8a) had an age of 8.26±0.54 ka, slightly older than the oldest Andisol age along the Reservoir fault (Sample #37, 7.47±0.08 ka), but also indicating soil formation after the LGM and that the moraine is likely a result of glaciation between 20 and 12 ka BP.

#### 6 Discussion




#### 395 6.1 Distributed deformation

Our analyses of the July 25, 2022  $M_w$ 5.7 EQ and geomorphic mapping and paleoseismic surveys in our study area indicate that there are several southwest–northeast striking active faults that accommodate right-lateral slip south of Chiles Volcano. The Reservoir fault and the InSAR analyses both show clear evidence of right-lateral Holocence fault ruptures. The best explanation for the linear scarps, landslides, and sag ponds along the Polylepis fault is also Holocene fault rupture, however further more detailed studies are required to prove this hypothesis. Nonetheless, right-lateral slip on these three parallel fault systems is consistent with the right-lateral strain at the northern boundary of the Quito-Latacunga microblock that is predicted by the Jarrin et al. (2023) geodetic block model (Fig. 1; Fig. 2b). Additionally, distributed strain across these three parallel fault systems, which are spread over  $\sim$ 10 km, is consistent with the lack of a sharp gradient in eastward velocities across the block boundary seen in InSAR (Fig.3 Marconato et al., 2024) and the distributed instrumental seismicity (Fig. 2b).

Based on our geomorphic mapping, a substantial portion of the 3 mm/yr of right-lateral strain is taken up by slip on the Reservoir fault. Considering the offset glacial moraines and creeks (Fig. 5), the ages of these features from our cosmogenic <sup>3</sup>He dating, and published ages we can estimate a slip rate on both the northern and southern strands of the fault. Assuming a maximum of 37 m of slip on both the northern and southern strands, and minimum age of 12 ka of the moraines (Table 1), we calculate a maximum slip rate of 3.1 mm/yr on both of these strands or a combined rate of 6.2 mm/yr. Given a minimum offset of 5 m on the northern strand, 12 m on the southern strand, and a maximum deglaciation age of 20.4 ka, we estimate a minimum slip rate of 0.2 and 0.6 mm/yr on the northern and southern strands respectively. The combined minimum slip rate is 0.8 mm/yr. Using a mean offset of 23 m for both the northern and southern strands, and an ice-free age of 15 ka (Rodbell et al., 2009; Blard et al., 2013; Angel et al., 2017), we obtain slip rates of 1.5 mm/yr for both strands and 3 mm/yr for the combined faults. Given these rates, slip on just the Reservoir fault could account for between one third to over double the total 3 mm/yr strain strain rate observed in the block model (Jarrin et al., 2023). Because we observe active deformation on the July 25 fault, and most likely on the Polylepis fault, we argue that deformation is distributed and that slip on the Reservoir fault takes up substantially less than 100% of the right-lateral strain at the north edge of the the Quito-Latacunga block. This




argument entails that either the minimum combined slip rate of the Reservoir fault (0.8 mm/yr) is more accurate than the higher rates (>6 mm/yr), or that the geodetic instantaneous strain rate is lesser than the geologic strain rate.

The minimum and mean slip rates we determine for the Reservoir fault are similar to right lateral slip rates determined on the northeast to east-northeast trending Buesaco (0.65 to 1.6 mm/yr) and Aranda (1.11 to 2.6 mm/yr) faults near Pasto, Colombia (Fig. 9). These active structures, which are located >50 km north of the block model boundary, show that the deformation zone may be even more spread out and distributed (e.g., ~70 km), or that it steps northward towards the east. Regional scale mapping (e.g., Veloza et al., 2012; Alvarado, 2012) also suggest numerous more northerly trending faults with a greater dipslip component within this region of distributed deformation. If these structures can be shown to be active, the north-stepping strike-slip faults shown in this study may act as tear faults in a eastward stepping northerly-trending thrust belt system (Fig. 9). However, field studies of these structures are required to determine if they are active and how they relate to the strike-slip faults mapped here. The Buesaco and Aranda faults are also proximal to the active Galeras volcano, similar to our study area's proximity to the Chiles-Cerro Negro volcano complex (Fig. 2), suggesting that the volcanic centers may be influencing where deformation is focused (e.g., Ebmeier et al., 2016; Ruch et al., 2016).

**Figure 9.** Active strike-slip faults mapped in this study and active stike-slip faults near Galeras with known slip rates. These faults are part of the proposed wide zone of distributed right-lateral deformation at the northern boundary of the Quito-Latacunga microblock (QL). The seemingly north-stepping right-lateral faults may act as tear faults in a east-stepping thrust fault system. Slip rates of the Reservoir fault (this study) and slip rates from the Buesaca and Aranda faults (Tibaldi and Leon, 2000) are shown as are the locations of active volcanoes with recent documented seismic swarms and inflation (Chiles-Cerro Negro and Galeras). RF: Reservoir fault, PF: Polylepis fault, JF: July 25 fault, BF: Buesaco fault, AF: Aranda fault.







# 6.2 Interaction with the Chiles-Cerro Negro Volcano complex

The proximity of our study area to recent inflation at the Chiles-Cerro Negro volcanic complex suggests that volcanism may not only localize deformation, but it also may account for a difference between geodetic and geologic slip rates. Uplift, and westward and eastward expansion away from a north-south axis centered on Chiles Volcano are visible in InSAR (Fig. 3), documenting volcanic inflation that has been undergoing since 2014 (Marconato et al., 2024). Ebmeier et al. (2016) used InSAR, GPS, and Coulomb stress modeling to propose that the 2014 right-lateral M 5.6 earthquake on a north- to northwest-striking fault was triggered by volcanic unrest and momentarily inhibited inflation. The July 25, 2022 right-lateral M 5.7 rupture along a east- to northeast-striking fault we document with InSAR (Fig. 4) probably also results from the volcanic inflation. The greatest eastward velocities in the regional InSAR data are just north of the July 25 fault (Fig. 3a). Inflation northwest of the fault that results in these velocities likely induces a horizontal stress in a northwest-southeast direction, which is compatible with the observed slip in the July 25 earthquake. The larger M 6.6 1868 earthquake may have also been related to volcanism at Chiles-Cerro Negro as volcanic activity was reported around that time (Monsalve and Laverde, 2016).

The influence that the Chiles-Cerro Negro volcano has on faults in the area may explain why combined right-lateral slip rates of the Reservoir, Polylepis, and July 25th faults could potentially be higher than geodetic slip rates. The calculated slip rate using mean offsets along the Reservoir fault alone is ~3 mm/yr, and given the evidence for recent activity on the other two faults it is very plausible that the combined geologic slip rate of these three faults is greater than the 3 mm/yr predicted in the Jarrin et al. (2023) block model. The geodetic block model uses GNSS velocities from a wide area across northwestern South America and southern Central America. A model that is consistent with all of these velocities may, by definition, ignore local high strain gradients like the one caused by inflation of the Chiles-Cerro Negro Volcano. Therefore, the faults in our study area may have greater earthquake rates and slip rates over the Holocene, resulting in part from volcanic inflation, than the geodetic block model predicts.

# **6.3** Reactivation of basement faults

The Polylepis and July 25 faults, and most of the lineations mapped in this study, are oriented oblique (more east-west) to the bedrock structures near Chiles-Cerro-Negro (Fig. 2b). The Reservoir fault, which trends northeast, however, may be more well aligned with some of these faults. Thick deposits of Quaternary volcanic and volcaniclastic rocks in the study area prevented any mapping of older bedrock faults in the immediate vicinity of these faults. But, based on the bedrock geology maps, bedrock structures south of Chiles are for the most part not being reactivated and are therefore not favorably aligned with the crustal stress conditions. This result is similar to studies at the southern edge of the Quito Latacunga microblock where Alvarado et al. (2016) show that ancient terrane bounding faults are not being reactivated by more recent and active faults. In contrast, along the eastern boundary of the microblock, active faults are parallel with and likely reactivate bedrock structures (Tibaldi et al., 2007; Alvarado et al., 2016). As mentioned previously, inflation of Chiles-Cerro Negro may change stress conditions at the northern edge of the Quito-Latacunga microblock, aligning them in a way that favors new fault formation. At the larger scale, the northern edge of the block is along strike from the main eastern boundary of the NAS and the northern edge of a large








constraining step-over of this boundary (Fig. 1). Therefore, the overall tectonic regime may also play a roles in changing stress conditions at this location.

## 6.4 Earthquake hazard

A slip rate, and earthquake recurrence and magnitudes are required to include crustal faults as seismic sources in probabilistic seismic hazard assessment (PSHA) models. For the Reservoir fault, we have determined slip rates above. Based on the earthquake history observed in the exposures of the Reservoir fault (Fig. 6 & Fig. 7) we can also estimate the recurrence interval for large surface rupturing earthquakes. Smaller earthquakes often don't rupture to surface, or may cause only small offset that may not be discernible in the stratigraphy (e.g., Canora et al., 2012). For example we did not observe surface offsets from the July 25 earthquake in this study, even though there was up to 8 cm of discrete offset observed with InSAR (Fig. 4). Even if there was surface offset, an offset of this magnitude in paramo environment would likely only offset Andisol against Andisol, making it impossible to perceive in trench or roadcut exposures. Therefore, we may only be observing the largest earthquakes in the exposures (e.g., M>6.5, Canora et al., 2012). The average time interval between earthquakes (or between the latest earthquake and present) we observe in the Reservoir fault outcrop (Fig. 7c) is  $2.3\pm0.5$  ka. However, given that the one of these intervals was calculated using the present-day as a minimum interval, the uncertainty is greater than we calculate.

Using fault scaling relations between overall rupture length and magnitude we can estimate magnitudes of the observed paleo-earthquakes. Considering ruptures of the entire estimated length of the northern branch of the Reservoir fault of 10.5 km to 25.5, the Wesnousky (2008) length scaling relation suggests a magnitude between M 6.3 and 6.7. Using a similar Leonard (2010) scaling relation, we estimate paleo-earthquake magnitudes between M 5.9 and M 6.5. A rupture of just the similarly ~10 km to 15 km long southern branch would also fall within this magnitude range. Given the observation of parallel coeval fault ruptures during the July 25 earthquake (Fig. 4d), and during other earthquakes (e.g., 2016 M 7.8 Kaikoura, Hamling et al., 2017; Morishita et al., 2017; Williams et al., 2018), we must consider that both segments have ruptured during a single event. If these faults were not connected at depth we would consider them as separate faults and consider their total summed length (20 km to 50 km) in a magnitude estimation. These lengths would predict paleo-earthquake magnitudes between M 6.6 and M 7.0 using the Wesnousky (2008) relation and between M 6.3 and M 6.9 using the Leonard (2010) relation.

We can also roughly estimate paleo-earthquake magnitudes using the offsets along the northern branch of the Reservoir fault in Fig. 5 assuming the earthquakes observed in Fig. 7 account for the majority of the displacement. If there have only been three large earthquakes since the LGM, and using a total offset between 5 m and 37 m of the lateral moraine, we estimate 1.7 m to 12.3 m of horizontal slip during each earthquake. With the Wesnousky (2008) relation between displacement and rupture length, rupture lengths would be between 28 km and 205 km corresponding to magnitudes of M 6.8 and M 7.7. Using the Leonard (2010) relation we estimate rupture lengths between 75 km and 830 km corresponding to magnitudes of M 7.2 and M 8.8. Only the 28 km rupture length estimation using the minimum average slip of 1.7 m and the Wesnousky (2008) scaling relation falls within the range of what we observe in the field (10 km to 50 km). We therefore exclude any estimates of magnitude using the Leonard (2010) relation. We also assume that if the total summed slip is greater than 5 m (three ~1.7 m earthquakes), it must represent earthquakes we do not observe in the fault exposure, or fault creep. These earthquakes may






be older than the oldest sediments offset in the exposures (8.54 to 8.39 ka), which is likely given the long time between deglaciation and the oldest event. The unaccounted for earthquakes could also have ruptured other fault segments (e.g., just the southern branch), not ruptured to surface, or have been smaller and not observable. In conclusion, based on the assumption of  $\sim$ 1.7 m of horizontal slip per earthquake, or 10.5 km to 50 km of rupture length, and using the Wesnousky (2008) scaling relation, we estimate that the paleo-earthquake magnitudes were likely between M 6.3 and M 7.0 (for either a single fault strand rupture or a coeval rupture of both strands).

We note there may be additional uncertainty resulting from shallow earthquake depths in the study area. The 2014 M 5.6 and the 2025 M 5.7 July 25 earthquake both have depths <5 km, and therefore may have a lesser rupture area for a given rupture length. The smaller rupture area would result in a lower magnitude and the magnitudes we determined above may be overestimated.

Our estimated paleo-earthquake magnitudes overlap with the proposed magnitude of the August 15, 1868 M 6.4 to M 6.8 earthquake that was determined to have a likely epicenter between El Angel and Chiles and close to the Reservoir, Polylepis, and July 25 faults (Fig. 2a, Beauval et al., 2010). We do not see evidence for a large earthquake as recent as 1868 in the exposure of the northern branch of the Reservoir fault, but this earthquake may have ruptured the southern branch of the Reservoir fault, or another structure in the area such as the Polylepis fault. The clear surface expression and recent landslides along the Polylepis fault (Fig. 8) leads us to speculate that it may have been the host of the 1868 earthquake, but detailed paleoseismic studies of this fault are needed to test this hypothesis.

## 515 7 Conclusion

Using remote sensing and field-based studies, we have been able to pinpoint where some of the right-lateral deformation at the northern boundary of the geodetically defined Quito-Latacunga microblock is accommodated by active faulting. InSAR analyses of a recent July 25, 2022 a M<sub>w</sub>5.7 right-lateral strike-slip earthquake, along with geomorphic mapping and field studies, show that the geodetically observed strain is accommodated by a >10 km wide zone of sub-parallel and discontinuous northeast-southwest trending faults, and the zone of distributed deformation may be much wider. These structures lack a coherent through-going geometry, are influenced by volcanism (notably Chiles-Cerro Negro), and show variable Holocene slip rates (0.8 to 6.2 mm/yr) that could cumulatively match or exceed the block model predictions. Seismicity and geodetic data further support this diffuse strain pattern, as neither seismic clusters nor GNSS velocity gradients align with a singular fault trace. Instead, the region resembles a crustal-scale distributed shear zone that transfers strain across a mechanically heterogeneous domain, modulated by volcanic, and lithospheric controls.

Volcanic inflation at Chile-Cerro Negro may increase earthquake recurrence and may result in geologic slip rates that exceed geodetic ones. The stress conditions imparted by volcanic inflation may also result in whether basement faults are reactivated at the northern boundary of the microblock. In the case of Chiles - Cerro Negro inflation, we observe that the basement structures are not being reactivated and new faults parallel with the geodetic block boundary have been formed. Given this observation, it is possible that active volcanism plays a part in where the microblock boundaries have formed.


The observed distributed deformation we observe still poses a substantial seismic hazard to communities in northern Ecuador and southern Colombia (e.g. El Angel and Pasto). Despite lesser fault lengths and significant segmentation, we show that these structures are capable of, and may have hosted the destructive the 1868 M 6.4 - 6.8 earthquake near El Angel. Further detailed geomorphic mapping and paleoseismic studies of other suspect structures along the northern boundary of the Quito-Latacunga microblock have the potential to find more active faults and further define seismic hazard in this region.

Code availability. Codes used for OxCal version 4.4.4 (Bronk Ramsey, 2021) and the shell script used to build a Digital Terrain Model (DTM) from the Pleiades satellite tri-stereo imagery using NASA Ames Stereoipeline (Beyer et al., 2018) are provided a data repository (Harrichhausen et al., 2025).

Data availability. A table showing cosmogenic sample locations and He isotope abundances, and a high resolution version of Figure 7 are provided a data repository (Harrichhausen et al., 2025).

Author contributions. Conceptualization: NH, LA, AA. Formal analysis: NH, LM. Funding acquisition: NH, LA, SB, HJ. Investigation: NH, LM, LA, PL, SB, HJ, AA, PHB. Methodology: NH, LM, JH, PHB. Project administration: NH, LA, AA. Resources: AA, IO. Supervision: LA, FR. Visualization: NH, LM. Writing (original draft preparation): NH, LM. Writing (review and editing): NH, LM, LA, PL, SB, HJ, AA, PHB, PM, FR, IO.

45 Competing interests. We declare that no competing interests are present.

Acknowledgements. This research was partly funded by a Centre National d'Études Spatiales (CNES) postdoctoral fellowship to N. Harrichhausen, and it benefited from the Institut de Recherche pour le Développement's (IRD) financial contribution to LMC14 (CEA-CNRS-IRD-IRSN-MCC). Additional funding for radiocarbon dating was provided by the Autorité de Sûreté Nucléaire et de Radioprotection (ASNR). A special thanks to J. Carcaillet and N. Paradis at ISTerre for their assistance with preparing samples for cosmogenic <sup>3</sup>He dating.

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
