# Peer review of "Distributed right-lateral strain at the northern boundary of the Quito-Latacunga microblock"

_EGUsphere, 2025_

## Referee Comment (RC1)

Review of: "Distributed right-lateral strain at the northern boundary of the Quito-Latacunga microblock" by Harrichhausen et al., submitted to Solid Earth

Reviewer: Dr Sam Wimpenny (University of Bristol, UK)

**Overview**

This manuscript presents new geomorphological and satellite remote-sensing observations of active faulting from the Inter-Andean Valley between Ecuador and Colombia. The authors use these new observations to test whether the distribution of strain inferred from the recently published block models of Jarrin et al., [2023] matches the kinematics and distribution of active faulting. They identify five new active faults across a ~70 km wide zone, which all lie north of an inferred block boundary. The authors also use paleoseismological trenching and Quaternary dating methods to determine the recent history of major earthquakes along two faults.

The manuscript is well-presented, with good quality figures and clear writing. I found it interesting to read and am sure others with specific interest in the tectonics and seismic hazard in the Northern Andes will find it a useful contribution to the literature.

I have some minor comments for the authors to consider before publication, but otherwise I recommend the article can be published after minor corrections.

**General Comments**

Consider emphasising the poor constraints of the block model boundary from the GNSS velocities: The Jarrin et al., [2013] block model boundary is constrained by three GNSS velocity measurements in the study region from what I can tell, and the model itself is not consistent with the velocities within uncertainty. The position of the block boundary is therefore poorly constrained. I would recommend highlighting this to motivate the manuscript, as where shear strain is being accommodated is poorly known. I say this because the argument that the authors make around finding the active faulting to be more distributed than predicted by the block model is expected – strain fields are almost always accommodated by multiple faults within continental lithosphere, particularly within mountain belts [McKenzie 1978].

Interpretation of distributed deformation in the InSAR data: I would recommend more clarity regarding what is meant by "distributed deformation" in Section 5.1. Some people use "distributed" to mean inelastic off-fault deformation [e.g. Milliner et al., 2025]. Here it is being used to mean any displacement around the fault that is not related to slip that reaches the surface. Such deformation can still reflect the elastic response of the wall rock to slip on parts of the fault buried at depth. Therefore, it is not necessarily converted into permanent deformation that would be visible in the geomorphological or geological record of the region (e.g. surface ruptures, subsidiary faults).

Consider adding some reflection on how precise the radiocarbon dates are given they are based on bulk sampling: The authors should consider: (1) describing what organic material think they were dating within the bulk radiocarbon samples, and (2) reflecting on what the true uncertainties may be on these stratigraphic dates that

are not captured by the formal lab-based uncertainties. Many studies assume a (conservative) ~1 kyr age uncertainty from bulk sampling because of the effects of bioturbation and root penetration meaning that organic matter does become mixed throughout the stratigraphy [e.g. Grutzner et al., 2016]. This mixing effect can explain why lots of trench sections have some ages that are out of stratigraphic order (just as the authors find in their trench and which they discount). However, the same processes will also affect other stratigraphic age estimates based on bulk sampling, even if it does not cause them to become out of stratigraphic order.

**Line-by-Line Comments**

- Line 28: Grammatical error "...whether deformation it is focused or...."
- Line 30: "Shear zones..." are particular structural geological structures in my eye maybe consider rephrasing to "further characterise the distribution of faulting that accommodates deformation in the upper crust"?
- Line 46: No comma in "with the recent rupture, reveals..."
- Line 48: Comma missing from "...faults lie within the proposed epicentral area, and could be responsible for, the 1868 M 6.8 ..."
- Line 48: Not sure this sentence makes sense to me: "By placing this study in context with ..." the study *is* the context as far as I can tell.
- Line 50: Would be worth stating quantitatively here what you consider "wide zone" to be? Is that 20 km or 200 km?
- Figure 1: Figure caption says "strain rates from Jarrin..." but you're showing slip rates across block boundaries and not strain rates.
- Line 71: Grammatical error "...faults along strike to the north-east that transverse..."
- Line 78: Consider rephrasing to: "...show 0.7-1.6 mm/yr and 1.1-2.6 mm/yr of right-lateral reverse slip across the Buesaco and Aranda Faults, respectively."
- Line 81: Consider rephrasing to "A southward decrease in the eastward component of the GNSS velocities across the northern boundary .... consistent with right-lateral shear strain on ENE-WSW striking planes predicted ..." more precise about the velocity gradients and relation to inferred strain field.
- Line 88: Worth being explicit that you're saying that the GNSS velocities with ~20 km of Chiles may capture transient volcanic deformation rather than velocities that are representative of the long-term tectonic deformation. As written, it could sound like *all* GNSS velocities may reflect volcanic deformation.
- Line 92: Consider rephrasing "... Colombia provides higher spatial resolution measurements of deformation aross the norther ..."

Line 94-95: You would only expect sharp velocity gradients across faults creeping near the surface or where there had been recent earthquakes. A gradient in the velocity over ~20 km is entirely consistent with there being elastic strain accumulation around a single fault which is locked in the top ~15-20 km of the crust [e.g. Wright et al., 2001], so could well be representative of a block boundary.

Line 98: Should be "analysis" not "analyses" as it's singular.

Line 105-107 and Figure 2: I would recommend providing the hypocentral locations of earthquakes alongside with the focal mechanisms of the larger events (Mw > 5) in Figure 2 if there are any. The focal mechanisms are key information as well for how the present-day strain field is being accommodated by faulting.

Line 115-119: Worth emphasising somehow that the earthquake was triggered, but that the total amount of strain generated by the episode of volcanic inflation was too small to account for the amplitude of fault slip. Therefore, there is probably tectonic strain accumulation in this area too, not just faulting entirely driven by magmatism. This means that the mechanism of the event should be related to the wider tectonic setting, rather than the local strain field caused by volcanic inflation/diking.

Section 3: Do you need this section? The information about Quaternary glaciation and landscape is important for understanding the sediments within which scarps are preserved, but could come in the Intro or section on fault-related geomorphology. The bedrock geology component seems overly detailed to me and the reader gets a little distracted here. I understand that later you compare the trends of the active faults with those within the bedrock geology, but that can simply be stated with citations later.

Line 171: Cite the filtering approach – has someone else tested this carefully? If it's new here, then it needs explaining in more detail. What type of filter are you using?

Line 172: Explain what unwrapping method you used? Did you not bother unwrapping the Sentinel-1 interferograms (why not)?

Line 185: Missing bracket closure around link.

Line 193-194: Were you specifically projecting the lateral moraine crests?

He-3 Cosmogenic Dating: Is sample erosion a factor in influencing the exposure age (this is not mentioned)? Do the authors account for this in the uncertainties on the estimated dates, or do they assume negligible erosion? Is there field evidence that supports this assumption?

Line 266: How many kilometres wide?

Line 273-274: Strikes should be quoted in 3 figures (060-070) as they're azimuths.

Line 292: Spelling error "...and then displaces the stream along ..."

Line 297: Correct grammar of this sentence: "Additionally, the undulating terrain relatively minor vertical ..."

Section 5.3: Looking at the terrain topography, have you tried creating structure contours for the fault to constrain its dip? It looks like it should be near vertical by the way it cross-cuts the topography in Figure 5b, but it is a simple exercise and would add evidence to support the inference that the faults are likely mostly strike-slip.

Figure 6 & 7 order: Figure 6 is only briefly mentioned before an extensive discussion of the observations in Figure 7 – consider switching the order in which these appear in the text so the reader doesn't jump backwards and forwards between figures.

Line 309: Should be "WSW-ENE-striking fault zone" based on Figure 5.

Line 311: Spelling "...narrows to a single strand..."

Line 338: Avoid starting sentence with "∼" – just use the word "Approximately".

Line 348: Grammar "... the the..."

Line 366: "Assume it is recycled" – more specifically you assume that the colluvial wedge contains sources of organic matter that are recycled and have been transported into a fracture. Could make this clearer in the text.

Section 6.2: It is unclear to me whether volcanic deformation would cause accelerated strike-slip faulting over the long term (i.e. ~10 kyrs) because it would require the strain from volcanic deformation be translated predominantly into permanent right-lateral strain on ENE-WSW trending planes. Volcanic deformation related to magma intrusions (dykes, sills, spherical magma reservoirs) induces dominantly vertical displacements [Okada et al., 1985; Yang et al., 1998], and therefore dominantly dipslip faulting [Rubin 1998]. Consider reflecting on whether this mechanism is mechanically feasible and consistent with the (absence of) evidence of dip-slip faulting.

Line 470: Change "don't" to "do not".

**References**

Wright, T., Parsons, B. and Fielding, E., 2001. Measurement of interseismic strain accumulation across the North Anatolian Fault by satellite radar interferometry. Geophysical Research Letters, 28(10), pp.2117-2120.

Grützner, C., Schneiderwind, S., Papanikolaou, I., Deligiannakis, G., Pallikarakis, A. and Reicherter, K., 2016. New constraints on extensional tectonics and seismic hazard in northern Attica, Greece: the case of the Milesi Fault. Geophysical Journal International, 204(1), pp.180-199.

Milliner, C., Avouac, J.P., Dolan, J.F. and Hollingsworth, J., 2025. Localization of inelastic strain with fault maturity and effects on earthquake characteristics. Nature Geoscience, pp.1-8.

McKenzie, D., 1978. Active tectonics of the Alpine—Himalayan belt: the Aegean Sea and surrounding regions. Geophysical Journal International, 55(1), pp.217-254.

---

## Referee Comment (RC2)

**Comments on "Distributed right-lateral strain at the northern boundary of the Quito-Latacunga microblock" by Harrichhausen et al (egusphere-2025-4329)**

The manuscript presents new geomorphological mapping, remote sensing, and dating results (radiocarbon and cosmogenic exposure ages) to investigate active faulting north of the Quito-Latacunga block (Northern Andes), offering valuable data that improves understanding of regional seismic hazard in a populated area. The authors provide neotectonic investigations and slip rate estimates for two faults, and they compellingly suggest that volcanic inflation from the Chiles-Cerro Negro volcanic complex may contribute to neotectonic activity by studying a recent 2023 earthquake. However, the study currently falls short of delivering a comprehensive analysis of the entire regional fault system, and several key morphotectonic features lack sufficient clarity or documentation to fully support the authors' regional assertions, ultimately restricting the scope and robustness of their final proposals. Please refer to my detailed comments below.

**I recommend that the article undergo major revisions prior to publication.**

**General comments**

- 1. Introduction: this section is well-structured. The only thing missing is more detail in the second paragraph (the state of the art). Please include more about the Neotectonics work done in Southern Colombia/Northern Ecuador over the last 20 years. (You already mention these papers later in the Tectonic Setting and the Discussion sections).
- 2. Tectonic Setting: My suggestion here is to make the material more accessible and easier to digest for the reader that is not familiar with the study area by using subsections to clearly delineate the topics covered (tectonic framework, GNSS, InSAR, instrumental and historical seismicity). For example, the recurring mention of historical seismicity in several paragraphs is redundant and should be streamlined.
- 3. Geologic Setting: The last two paragraphs discussing volcanic and glacial activity are too long and should be simplified and made more concise.
  Importantly, the manuscript omits discussion of the Romeral Shear Zone (RSZ) (or Romeral Fault System). This system is considered a reactivated suture zone with an origin distinct from the Cayambe-Afiladores-Sibundoy fault system. Since the RSZ is active in the study area through structures like the Buesaco and Aranda Faults. The authors should be clearer about this system and its neotectonics implications throughout the paper.
- **4.** Methods: As with the Tectonic Setting section, I would recommend the authors split this section into subsections given the numerous techniques they used. Also, more details about cosmogenic dating should be provided (See comments below).
- 5. Results: The structure of this section is generally good, but the following points require revision:
  - a. I have difficulty identifying the claimed offset moraines or channels based on the DTM and external imagery. Please provide more detailed information on the topography analysis to validate these estimates.
  - b. To reflect the dating of multiple structures, move the Dating subsection outside of the Reservoir Fault section.
  - c. Include additional details on the Polylepis Fault and the surrounding lineaments, as these features are important for future structural interpretations and neotectonic works.

- **6.** Discussion: This section lacks an analysis of all the different fault systems present in northern Ecuador and southern Colombia. For example, some of the conclusions about the role of the reactivated suture zones (like the Romeral Fault System) are simplified by citing a paper that describe the tectonics of southern Ecuador, far from the study area.
  - a. The authors propose volcanic inflation as a potential conditioning factor (or triggering mechanism) for the occurrence of the July 2022 earthquake. Nevertheless, no further explanation is provided. A schematic figure would be of great help to strengthen your hypothesis.

**Detailed comments line by line**

Line 15. The right name is Chiles-Cerro Negro volcanic system (there is two volcanoes in this system: Chiles and Cerro Negro). Please revise the manuscript to ensure consistent usage of Chiles-Cerro Negro Volcanic System (or CCN-VS if you prefer) throughout. For example, in line 153 you refer to it as the Cerro Negro-Chiles Volcano while in line 157 is called the Chiles-Cerro Negro volcanic complex.

Line 28. Remove "it" after "... whether deformation..."

Line 34. Can you be more specific on these boundaries?

Line 40. Please make sure the epicenter of the July 25, 2022, earthquake is added to Figure 1. This location is crucial as it highlights part of the motivation for the study and represents the source area for the InSAR analysis you present.

Line 60 ("These studies highlight...") feels out of place or disconnected from the preceding text. You attempt to emphasize your motivation, but you should use a better transition or connector at the start of the sentence to improve the logical flow of the paragraph.

Lines 62-63 Please rephrase to be more concise.

Lines 79-80 The area between Ibarra (Ecuador) and Pasto (Colombia) has experienced several historic earthquakes. Please consult the historic seismicity project led by the Colombian Geological Survey (https://sish.sgc.gov.co/visor/). It would be useful to add these additional historic events to Figure 2.

Line 93. Why is it important or not to identify a sharp gradient in velocities? What would be the implications for the seismic hazard?

Line 121. Make sure Inter-Andean Valley is consistent throughout the paper (you used Interandean valley in line 6 or Inter Andean Valley in line 129)

Lines 128-132. You mentioned three oceanic plateaus but only two terranes: San Juan and Guaranda. Is there a terrane missing? Were not all of them accreted?

Lines 137-138. This tectonic interpretation may be valid for Ecuador, but the situation in southern Colombia is complicated by the active suture zones. These zones display extensive evidence (geomorphological, historical, and instrumental) of recent activity, resulting in complex interaction with the NE-SW structures. The Romeral Shear Zone (or Romeral Fault) is the best illustration of this complexity (see Ego et al., 1995; Paris et al., 2000; Vinasco, 2019; Garcia-Delgado et al., 2022). The Buesaco and Aranda faults that are mentioned in lines 77-78 for the first time are part of the Romeral system and represent reactivated segments of the sutures zones between oceanic and continental blocks (see Paris et al., 2000; Tibaldi and Romero, 2000).

Line 144. Remove the extra parenthesis when calling Fig 2.

Line 176. Please Correct Colmbia.

Line 177. The provided URL leads to the webpage of the Colombian Geological Survey, not the specific data repository for the DTM. Please provide a direct link to the data service or repository so that the metadata can be properly verified.

Lines 196-199. Was this issue related to the way the Pleiades DTM is modeled? Were the moraines too small?

Lines 203-206 The description of the exposure dating techniques is insufficient, especially considering their importance in supporting the manuscript's interpretations. Please expand this section to provide details, including the fundamental assumptions underlying the exposure dating method (e.g., nuclide production rates (spallogenic and muonic), shielding, erosion history). Specifically, the authors must clearly outline the assumptions, the mathematical formulations, and the sources and magnitude of uncertainties behind the reported ages.

Lines 205-206. Did you consider complex burial/re-exposure history? What field evidence do you have to reject this scenario?

Lines 219-220. Please provide more details about this assumption. What corrections are needed if all measured He4 is magmatic in origin?

Lines 284-294. The offset moraines and channels mentioned are not visually apparent in Figure 5 or verifiable using independent satellite imagery (like Google Earth Pro). Please provide more detailed information on the methodology used to project the moraine ridges onto Figure 5c so that these features can be verified.

Figure 5. Can you color the slope maps in panels C and D?

Figure 7. This is a really nice Figure!

Line 375 Grammar "... of the of the fault exposures..."

Section 5.4 After carefully inspecting Google Earth, one can notice that the area of the Polylepis Fault is made by more than one structure that would be worth including in your Figure 8. The new imagery provides a better contrast to aim in the identification of the fault traces.

Satellite image from Google Earth (image © CNES / Airbus)

Line 399. Remove "...more..." to avoid redundancy

Line 400. Why are these structures considered fault systems? Also, uou mention three fault systems but only two faults: Polylepis and Reservoir.

Line 403. Shouldn't the two faults be one fault system if that's the case?

Line 404. How did you define the width of the fault zone?

Line 405. What do you mean by right-lateral strain?

Lines 406-407. As mentioned above, please consider my comments on the offset features and the uncertainty around the interpretation of recent displacements along the Reservoir Fault.

Line 416. This is the first time you mention the July 25 fault in the main body of the manuscript. Please ensure the fault is named and described consistently upon its first relevant mention. Furthermore, I suggest replacing this name with a geographically significant name to facilitate referencing.

Line 418. You didn't calculate strain rate but slip rates. Please review the article to avoid mixing concepts such as slip rate and strain rate, which are not the same.

Lines 420-423. An examination of the mapped structures and your interpretation (Fig. 2) suggests a plausible southern extension of the Romeral Shear Zone (Buesaco and Aranda faults). Specifically, lineaments southwest of the Galeras volcano (potentially belonging to the Cauca-Patía system?) appear to show a southward prolongation, possibly merging with the Reservoir and Polylepis Faults. Given that both the Cauca-Patía and Romeral structures are right-lateral faults, their regional kinematics align well with the mapped structures in the study area.

If this is true, it poses a significant tectonic implications: this extended zone may delineate either a separate, unmapped microblock or represent the northern boundary/extension of the QL microblock.

Lines 428-430. This interpretation is out of place because it hasn't been developed before. Only in section 6.2 you discuss the potential influence of the CCN-VC.

Lines 437-443. The role of volcanic inflation should be more elaborated since it's a key conditioning factor for the triggering of the July earthquake. It'd be helpful to see that on a map or a new schematic figure.

Lines 441-442. I'd recommend to remove this interpretation about the 1868 earthquake since its speculative.

Line 456-458. The presence of thick volcaniclastic deposits inherently presents significant difficulty in observing recent faulting. Consequently, the lack of clear surface expression through the sedimentary cover does not preclude the underlying structure from being active. For example, you didn't observe a rupture for the July 25 earthquake. So the same reasoning should be valid for the N-NE suture zones.

Line 458. Please refer to my recommendation and include in your analysis the Romeral and Cauca-Patia fault systems which in southern Colombia are considered tectonically active. The Buesaco and Aranda faults are part of the Romeral fault system (or shear zone depending on the authors).

Lines 462-465. Please rewrite this section as the current discussion is unclear. Furthermore, to support the proposed earthquake conditioning factors, please add a figure showing the modeled horizontal stress field from the CCN-Vc and its orientation relative to your mapped fault structures.

Line 470 Avoid contractions (don't)

Lines 488-494. I'd recommend to keep only the radiocarbon-based paleoearthquake estimates for the Reservoir Fault. The moraine and cosmogenic 3He estimates are too vague; their 2σ uncertainty (2 kyr) is too large to constrain the recurrence interval defined by the radiocarbon dating.

**References**

Ego, F., Sebrier, M., Yepes, H., 1995, Is the Cauca-Patia and Romeral fault system left of rightlateral? Geophysical Research Letters, v. 22, p. 33-36.

Garcia-Delgado, H., Velandia, F., Bermudez, M., Audemard, F., 2022. The present-day tectonic regimes of the Colombian Andes and the role of slab geometry in intraplate seismicity. International Journal of Earth Sciences, v. 111, p. 2081-2099.

París, G., Machette, M.N., Dart, R.L., Haller, K.M., 2000, Map and Database of Quaternary Faults and Folds in Colombia and Its Offshore Regions. Open-File Report 2000–284. USGS.

Tibaldi, A., And Romero, J., 2000, Morphometry of late Pleistocene-Holocene faulting and volcanotectonic relationship in the southern Andes of Colombia. Tectonics, v. 19, p. 358-377.

Vinasco, C., 2019, The Romeral shear zone. In: Cediel F, Shaw R (eds) Geology and tectonics of Northwestern South America: the Pacific-Caribbean-Andean junction. Front Earth Sci 833–876. https://doi. org/10.1007/978-3-319-76132-9 12

---

## Author Comment (AC1)

Dr. Nicolas Harrichhausen
Assistant Professor of Geology
Dept. Geological Sciences
njharrichhausen@alaska.edu

We thank the Dr. Sam Wimpenny for their thoughtful comments and suggestions for our manuscript " Distributed right-lateral strain at the northern boundary of the Quito-Latacunga microblock" that we submitted for publication in *Solid Earth.* This letter contains our responses to the comments and suggestions by the reviewer. We paraphrase the reviewer's comments in bold and follow with our responses and descriptions of relevant edits to a revised manuscript that will be resubmitted for review.

Review by Dr. Sam Wimpenny:

**Consider emphasizing the poor constraints of the block model boundary from the GNSS velocities...I would recommend highlighting this to motivate the manuscript, as where shear strain is being accommodated is poorly known. ...– strain fields are almost always accommodated by multiple faults within continental lithosphere, particularly within mountain belts [McKenzie 1978].**

Thank you for this suggestion. We agree that not only are the structures that accommodate deformation not well understood, that the actual boundary location itself is not well constrained. We have edited the following sentence in the introduction to emphasize this point: "However, the location of the northern edge of the Quito-Latacunga microblock and any structures that accommodate deformation predicted here have not been well constrained, especially at the scale of this study."

We also note the uncertainty in the boundary location due to the low density of GNSS stations in the study area in the Tectonic setting section.

**Interpretation of distributed deformation in the InSAR data: I would recommend more clarity regarding what is meant by "distributed deformation" in Section 5.1. Some people use "distributed" to mean inelastic off-fault deformation [e.g. Milliner et al., 2025]. Here it is being used to mean any displacement around the fault that is not related to slip that reaches the surface**.

We agree with the need to clarify what we mean by distributed deformation in this section and we have added the following sentence to do so:

"This 13 cm of total surface deformation includes any displacement along the fault that did not rupture the surface, and/or inelastic, off-fault, deformation in the area around the fault".

Additionally, we have checked all instances where we mention distributed deformation and ensured it is clear that we are referring to deformation across several fault zones.

**Consider adding some reflection on how precise the radiocarbon dates are given they are based on bulk sampling: The authors should consider: (1) describing what organic material think they were dating within the bulk radiocarbon samples, and (2) reflecting on what the true uncertainties may be on these stratigraphic dates that are not captured by the formal lab-based uncertainties. Many studies assume a (conservative) ~1 kyr age uncertainty from bulk sampling because of the effects of bioturbation and root penetration meaning that organic matter does become mixed throughout the stratigraphy [e.g. Grutzner et al., 2016]...**

We thank the reviewer for this constructive point and have added the following text in the Methods to discuss the uncertainty in our sampling strategy some more.

"The majority of our samples were collected from Páramo soils (Andisol), which have very high organic carbon content. Their low bulk density, high porosity, and humic/dark nature make them comparable to peat-like soils or organic-rich soils. Because of these properties, small-volume, focused sampling can retrieve sufficient organic carbon for radiocarbon dating, reducing the risk of time-averaging or mixing compared to bulk sampling in low-organic, mineral sediments. The high organic content and relative stability of organic matter in Andisol under cold, humid conditions at ~4000 m reduces decomposition and thus preserves carbon, making them good candidates for radiocarbon dating. These types of samples have been shown to provide reliable radiocarbon ages in previous paleoseismic studies in similar environments along the Pallatanga fault zone (Baize et al., 2015, 2020) and the Billecocha fault system (Jomard et al., 2021) An identical sampling strategy resulted in precise historical ages matching with the 1797 earthquake in Pallatanga area (Baize et al., 2015). Nevertheless, we acknowledge that post-depositional processes such as bioturbation and root penetration could introduce additional uncertainty that is not fully captured by the laboratory-reported analytical errors (e.g., Grützner, 2015). Additionally, our samples could be detrital giving ages that are older than the unit they are located within. This is especially the case for samples from colluvial wedges (e.g., DuRoss et al., 2022). Therefore, we assessed each sample and date and discounted dates that were likely to be affected by bioturbation and re-sedimentation.

Finally, we note that although our radiocarbon ages do not have an arbitrary uncertainty imposed on them as suggested, the units they are dating all have age ranges greater than 1000 years. Additionally, our earthquake event ages span ~800, 1500, and 1400 years. Thus, while we have kept the analytical uncertainty, we believe our method of interpretation has not overly constrained the unit or earthquakes ages.

*Line-by-Line Comments*
**Line 28: Grammatical error "…whether deformation it is focused or…."**
Edited.

**Line 30: "Shear zones…" are particular structural geological structures in my eye – maybe consider rephrasing to "further characterise the distribution of faulting that accommodates deformation in the upper crust"?**
We have made the suggested edit.

**Line 46: No comma in "with the recent rupture, reveals…"**
We have made the suggested edit.

**Line 48: Comma missing from "…faults lie within the proposed epicentral area, and could be responsible for, the 1868 M 6.8 …"**
We have removed all commas.

**Line 48: Not sure this sentence makes sense to me: "By placing this study in context with …" – the study *is* the context as far as I can tell.**
We have changed "this study" to "these results" to clarify the sentence and have cited the "parallel faults in Colombia" to emphasize that these are based on previous results.

**Line 50: Would be worth stating quantitatively here what you consider "wide zone" to be? Is that 20 km or 200 km?**
We have added "an ~70-km-wide zone"

**Figure 1: Figure caption says "strain rates from Jarrin…" but you're showing slip rates across block boundaries and not strain rates.**
We have changed "strain rates" to "slip rates"

**Line 71: Grammatical error "…faults along strike to the north-east that transverse…"**
Northeast is not hyphenated in American English, which we are using in this manuscript.

**Line 78: Consider rephrasing to: "…show 0.7-1.6 mm/yr and 1.1-2.6 mm/yr of right-lateral reverse slip across the Buesaco and Aranda Faults, respectively."**
We have made the suggested edit.

**Line 81: Consider rephrasing to "A southward decrease in the eastward component of the GNSS velocities across the northern boundary …. consistent with right-lateral shear strain on ENE-WSW striking planes predicted …" – more precise about the velocity gradients and relation to inferred strain field.**
We have made the suggested edit.

**Line 88: Worth being explicit that you're saying that the GNSS velocities with ~20 km of Chiles may capture transient volcanic deformation rather than velocities that are representative of the long-term tectonic deformation. As written, it could sound like *all* GNSS velocities may reflect volcanic deformation.**
We have made the suggested edit.

**Line 92: Consider rephrasing "… Colombia provides higher spatial resolution measurements of deformation aross the norther …"Line 94-95: You would only expect sharp velocity gradients across faults creeping near the surface or where there had been recent earthquakes. A gradient in the velocity over ~20 km is entirely consistent with there being elastic strain accumulation around a single fault which is locked in the top ~15-20 km of the crust [e.g. Wright et al., 2001], so could well be representative of a block boundary.**
Thank you for bringing this to our attention. The width over which the InSAR shows a reduction in velocities is over ~60 km, not 20 km, so we have corrected that.

**Line 98: Should be "analysis" not "analyses" as it's singular.**
We have made the suggested edit.

**Line 105-107 and Figure 2: I would recommend providing the hypocentral locations of earthquakes alongside with the focal mechanisms of the larger events (Mw > 5) in Figure 2 if there are any. The focal mechanisms are key information as well for how the present-day strain field is being accommodated by faulting.**

Good suggestion. We have added focal mechanisms for all EQs (Mw>4.5) since 1980 in the map area. Some have a non-double couple component indicating volcanic origins.

**Line 115-119: Worth emphasising somehow that the earthquake was triggered, but that the total amount of strain generated by the episode of volcanic inflation was too small to account for the amplitude of fault slip. Therefore, there is probably tectonic strain accumulation in this area too, not just faulting entirely driven by magmatism. This means that the mechanism of the event should be related to the wider tectonic setting, rather than the local strain field caused by volcanic inflation/diking.**

Thank you for this suggestion. We have added the following text to emphasize the potential tectonic origin of the stress: "InSAR and Coulomb stress modeling suggest that this earthquake could have been induced by inflation south of the volcano that could be attributed to volcanic inflation and a pore fluid pressure increase. However, its focal mechanism is consistent with tectonic stress and volcanism could potentially be just a trigger of the earthquake."

**Section 3: Do you need this section? The information about Quaternary glaciation and landscape is important for understanding the sediments within which scarps are preserved, but could come in the Intro or section on fault-related geomorphology. The bedrock geology component seems overly detailed to me and the reader gets a little distracted here. I understand that later you compare the trends of the active faults with those within the bedrock geology, but that can simply be stated with citations later.**

The relationships between active faulting, glacial geomorphology, volcanism, and inherited bedrock structures are discussed in detail in the Discussion of our manuscript. We feel that this information is essential to introduce and this section is the place to do so. The second reviewer suggested more detail in the bedrock geology section, but shortening of glacial and volcanic activity paragraphs, and we chose to follow their advice as it was one of their main comments.

**Line 171: Cite the filtering approach – has someone else tested this carefully? If it's new here, then it needs explaining in more detail. What type of filter are you using?**

This filtering approach has been described and tested in Marconato et al., 2024 (https://doi.org/10.1016/j.srs.2023.100113). We filter the interferogram with a standard sliding median, in order to smooth out part of the strong phase gradients resulting from earthquake displacements. Removing the gradient in the sliding window prior to filtering and reintroducing it after helps preserve the coseismic fringes.

We have added the Marconato et al., 2024 reference to our manuscript.

**Line 172: Explain what unwrapping method you used? Did you not bother unwrapping the Sentinel-1 interferograms (why not)?**

We used a region growing algorithm (we have added this text to the manuscript).

We unwrapped the Sentinel-1 interferogram as well but the unwrapped ALOS-2 is cleaner. We have now included two unwrapped Sentinel-1 interferograms in the data supplement.

**Line 185: Missing bracket closure around link.**

Added bracket.

**Line 193-194: Were you specifically projecting the lateral moraine crests?**
Yes we were specifically projecting the moraine crests, which were mapped based on hillshades, slope maps, and topographic contours. We now provide some more detail on this methodology in the methods section.

**He-3 Cosmogenic Dating: Is sample erosion a factor in influencing the exposure age (this is not mentioned)? Do the authors account for this in the uncertainties on the estimated dates, or do they assume negligible erosion? Is there field evidence that supports this assumption?**
The cold climate and unaltered appearance of the basalt boulders suggest little chemical weathering allowing us to assume that sample erosion is negligible. We now state this in the manuscript.

**Line 266: How many kilometres wide?**
Thank you for catching the typo. It should read ~15 km-wide.

**Line 273-274: Strikes should be quoted in 3 figures (060-070) as they're azimuths.**
We have made the suggested edit.

**Line 292: Spelling error "…and then displaces the stream along …"**
Good catch!

**Line 297: Correct grammar of this sentence: "Additionally, the undulating terrain relatively minor vertical …"**
Edited.

**Section 5.3: Looking at the terrain topography, have you tried creating structure contours for the fault to constrain its dip? It looks like it should be near vertical by the way it cross-cuts the topography in Figure 5b, but it is a simple exercise and would add evidence to support the inference that the faults are likely mostly strike-slip.**
Yes, we did attempt to construct contours at locations where the fault crosses the glacial valleys. The individual fault segments here are subvertical, or dip steeply north suggesting strike-slip structures.

**Figure 6 & 7 order: Figure 6 is only briefly mentioned before an extensive discussion of the observations in Figure 7 – consider switching the order in which these appear in the text so the reader doesn't jump backwards and forwards between figures.**
This is a good suggestion but we believe that Figure 6 is the best place to fit in the scarp photo, and it should be presented before the outcrops.We have switched the order of the outcrop descriptions so the outcrop in Figure 6 is described first and therefore is closer to the figure in the paper.

**Line 309: Should be "WSW-ENE-striking fault zone" based on Figure 5.**
Thank you for catching that typo.

**Line 311: Spelling "…narrows to a single strand…"**
Edited.

**Line 338: Avoid starting sentence with "~" – just use the word "Approximately".**
Edited.

**Line 348: Grammar "… the the…"**
Edited.

**Line 366: "Assume it is recycled" – more specifically you assume that the colluvial wedge contains sources of organic matter that are recycled and have been transported into a fracture. Could make this clearer in the text.**
Thank you for your suggestion. We have changed the sentence to read: "Because of this age discrepancy and the observation that #39 is part of a colluvial wedge (CW2), we assume that this sample has been recycled and has been transported into a fracture."

**Section 6.2: It is unclear to me whether volcanic deformation would cause accelerated strike-slip faulting over the long term (i.e. ~10 kyrs) because it would require the strain from volcanic deformation be translated predominantly into permanent right-lateral strain on ENE-WSW trending planes. Volcanic deformation related to magma intrusions (dykes, sills, spherical magma reservoirs) induces dominantly vertical displacements [Okada et al., 1985; Yang et al., 1998], and therefore dominantly dip- slip faulting [Rubin 1998]. Consider reflecting on whether this mechanism is mechanically feasible and consistent with the (absence of) evidence of dip-slip faulting.**
Thank you for making this point. We agree that it likely wouldn't increase long-term slip rates, but volcanic unrest may increase the short-term slip rate due to an increased number of earthquakes during volcanic activity.

Given the shallow inflation (as evidenced by shallow seismicity) and the clear westward and eastward crustal motion away from an area immediately south of Chiles, we feel that strike-slip rupture would be mechanically feasible. Volcanic inflation has been shown to induce strike-slip earthquakes at the Makushkin volcano in Alaska (Chang and Grapethin, 2024). It is beyond the scope of this manuscript to test our suggested trigger with a rigorous coulomb stress model, but this would be a great follow up study. We have added text to section 6.2 to explain this and have referenced Chang and Grapethin (2024).

**Line 470: Change "don't" to "do not".**
Edited

We thank Dr. Wimpenny for their comments and suggestions, we believe they have greatly strengthened the manuscript.

Nicolas Harrichhausen et al.

---

## Author Comment (AC2)

Dr. Nicolas Harrichhausen
Assistant Professor of Geology
Dept. Geological Sciences
njharrichhausen@alaska.edu

We thank the reviewer for their thoughtful comments and suggestions for our manuscript " Distributed right-lateral strain at the northern boundary of the Quito-Latacunga microblock" that we submitted for publication in *Solid Earth.* This letter contains our responses to the comments and suggestions by Reviewer 2. We paraphrase their comments in bold and follow with our responses and descriptions of relevant edits to a revised manuscript that will be resubmitted for review.

**Introduction: this section is well-structured. The only thing missing is more detail in the second paragraph (the state of the art). Please include more about the Neotectonics work done in Southern Colombia/Northern Ecuador over the last 20 years. (You already mention these papers later in the Tectonic Setting and the Discussion sections).**
Thank you for your suggestion. We have changed the sentence in the second paragraph indicating that while active faults have been confirmed in southern Colombia, they are not aligned with or that close to the predicted block model boundary.

**Tectonic Setting: My suggestion here is to make the material more accessible and easier to digest for the reader that is not familiar with the study area by using subsections to clearly delineate the topics covered (tectonic framework, GNSS, InSAR, instrumental and historical seismicity). For example, the recurring mention of historical seismicity in several paragraphs is redundant and should be streamlined.**
We have added subtitles to the sections in the Tectonic Setting section. We are not sure as to what the reviewer is referring to by "recurring mention of historical seismicity" as it is only mentioned in the last two paragraphs (both of which discuss seismicity).

**Geologic Setting: The last two paragraphs discussing volcanic and glacial activity are too long and should be simplified and made more concise. Importantly, the manuscript omits discussion of the Romeral Shear Zone (RSZ) (or Romeral Fault System). This system is considered a reactivated suture zone with an origin distinct from the Cayambe-Afiladores-Sibundoy fault system. Since the RSZ is active in the study area through structures like the Buesaco and Aranda Faults. The authors should be clearer about this system and its neotectonics implications throughout the paper.**
Thank you for pointing out the Romeral fault zone, we have now included text clarifying its significance as a reactivated structure in the region.

We have also shortened the volcanic and glacial activity section by removing some of the details. The glacial geomorphology section is essential as we are using glacial geomorphologic markers to constrain our slip rates, thus we hesitate to shorten these sections too much.

**Methods: As with the Tectonic Setting section, I would recommend the authors split this section into subsections given the numerous techniques they used. Also, more details about cosmogenic dating should be provided (See comments below).**
We have added subtitles. See reply to the comments on cosmogenic dating below.

**Results: The structure of this section is generally good, but the following points require revision:**

**I have difficulty identifying the claimed offset moraines or channels based on the DTM  and external imagery. Please provide more detailed information on the topography analysis to validate these estimates.**
See reply on the offset moraine identification comments below.

**To reflect the dating of multiple structures, move the Dating subsection outside of the Reservoir Fault section.**
The dating subsection in the Reservoir fault section only discusses dating of the Reservoir fault. Dating of the Polylepis fault is within that section. We have clarified this by adding a "Dating" subsection to the Polylepis fault results.

**Include additional details on the Polylepis Fault and the surrounding lineaments, as these features are important for future structural interpretations and neotectonic works.**
We have added the new lineament suggested by the reviewer to Figure 8 and have described it in the Polylepis fault section. See reply to comments on the Polylepis fault below for more details.

**Discussion: This section lacks an analysis of all the different fault systems present in northern Ecuador and southern Colombia. For example, some of the conclusions about the role of the reactivated suture zones (like the Romeral Fault System) are simplified by citing a paper that describe the tectonics of southern Ecuador, far from the study area.**
We agree that more introductory material and discussion is needed on the Romeral fault zone in southern Colombia and have added text that places it in context with our study. The cited paper discussed faulting in Central Ecuador along the eastern boundary of the Quito-Latacunga microblock, which is specifically mentioned as a comparison to our study area. We have edited the text to clarify where this study took place.

**The authors propose volcanic inflation as a potential conditioning factor (or triggering mechanism) for the occurrence of the July 2022 earthquake. Nevertheless, no further explanation is provided. A schematic figure would be of great help to strengthen your hypothesis.**
This is a good suggestion, however upon imagining a schematic figure we feel that the regional InSAR figure (Figure 3a) is a great reference for our proposed earthquake triggering. We have therefore explained our mechanism further with the following text referring to Figure 3a and have added arrows showing the inflation direction to Figure 3a. These arrows should assist the reader in visualizing our proposed earthquake mechanism.

**Detailed comments line by line**

**Line 15. The right name is Chiles-Cerro Negro volcanic system (there is two volcanoes in this system: Chiles and Cerro Negro). Please revise the manuscript to ensure consistent usage of Chiles-Cerro Negro**

**Volcanic System (or CCN-VS if you prefer) throughout. For example, in line 153 you refer to it as the Cerro Negro-Chiles Volcano while in line 157 is called the Chiles-Cerro Negro volcanic complex.**

Thank you for pointing out this inconsistency, we have edited the term throughout the manuscript.

**Line 28. Remove "it" after "...whether deformation..." Line 34. Can you be more specific on these boundaries?**

We have removed "it". We are unclear one what you mean by more specific. The locations and the faults that define the boundaries are described and shown in Figure 1.

**Line 40. Please make sure the epicenter of the July 25, 2022, earthquake is added to Figure 1. This location is crucial as it highlights part of the motivation for the study and represents the source area for the InSAR analysis you present.**

We have added the July 22 EQ epicenter to Figure 1.

**Line 60 ("These studies highlight...") feels out of place or disconnected from the preceding text. You attempt to emphasize your motivation, but you should use a better transition or connector at the start of the sentence to improve the logical flow of the paragraph.**

We respectfully disagree as the studies in the previous sentence are precisely what we are referring to in this sentence.

**Lines 62-63 Please rephrase to be more concise.**

We believe this sentence cannot be made more concise without losing its meaning.

**Lines 79-80 The area between Ibarra (Ecuador) and Pasto (Colombia) has experienced several historic earthquakes. Please consult the historic seismicity project led by the Colombian Geological Survey (https://sish.sgc.gov.co/visor/). It would be useful to add these additional historic events to Figure 2.**

The earthquakes we believe the reviewer is referring to are mentioned in the paragraph on instrumental and historical seismicity.

**Line 93. Why is it important or not to identify a sharp gradient in velocities? What would be the implications for the seismic hazard?**

A sharper gradient in velocities would suggest more localized crustal deformation. More strain localization could suggest a single throughgoing fault where all the slip is concentrated. Having a single fault versus several faults would affect how seismic hazard is modeled. We have edited the text to explain this point more clearly.

**Line 121. Make sure Inter-Andean Valley is consistent throughout the paper (you used Interandean valley in line 6 or Inter Andean Valley in line 129)**

Thank you for catching this. We have made edits to be consistent throughout the manuscript.

**Lines 128-132. You mentioned three oceanic plateaus but only two terranes: San Juan and Guaranda. Is there a terrane missing? Were not all of them accreted?**

There were three terranes accreted but only two are exposed in our study area. We only provided the names of those two terranes.

**Lines 137-138. This tectonic interpretation may be valid for Ecuador, but the situation in southern Colombia is complicated by the active suture zones. These zones display extensive evidence (geomorphological, historical, and instrumental) of recent activity, resulting in complex interaction with**

the NE-SW structures. The Romeral Shear Zone (or Romeral Fault) is the best illustration of this complexity (see Ego et al., 1995; Paris et al., 2000; Vinasco, 2019; Garcia-Delgado et al., 2022). The Buesaco and Aranda faults that are mentioned in lines 77-78 for the first time are part of the Romeral system and represent reactivated segments of the sutures zones between oceanic and continental blocks (see Paris et al., 2000; Tibaldi and Romero, 2000).

See response to comments above on the Romeral fault system.

**Line 144. Remove the extra parenthesis when calling Fig 2. Line 176. Please Correct Colmbia.**

We have made the suggested edits.

**Line 177. The provided URL leads to the webpage of the Colombian Geological Survey, not the specific data repository for the DTM. Please provide a direct link to the data service or repository so that the metadata can be properly verified.**

We have now provided the DTM in the supplemental data repository linked to this manuscript.

**Lines 196-199. Was this issue related to the way the Pleiades DTM is modeled? Were the moraines too small?**

This issue is unrelated to the DTM, as it is a function of the large width, subtle crests, and undulating topography of the lateral moraines (as mentioned at the end of the paragraph).

**Lines 203-206 The description of the exposure dating techniques is insufficient, especially considering their importance in supporting the manuscript's interpretations. Please expand this section to provide details, including the fundamental assumptions underlying the exposure dating method (e.g., nuclide production rates (spallogenic and muonic), shielding, erosion history). Specifically, the authors must clearly outline the assumptions, the mathematical formulations, and the sources and magnitude of uncertainties behind the reported ages.**

The cited papers in the methods section provide the requested details about the dating technique and we believe that rewriting these methods would be redundant.

**Lines 205-206. Did you consider complex burial/re-exposure history? What field evidence do you have to reject this scenario?**

This is a good point. The selected samples were fresh with little to no alteration therefore we made the assumption that there was little to no exposure before weathering and erosion. We now state this assumption in the methods section. However, a complex burial/re-exposure history is still very much possible, and may explain the very old age of the boulder on the moraine near the Polylepis fault, which we state in the Polylepis dating section.

**Lines 219-220. Please provide more details about this assumption. What corrections are needed if all measured He4 is magmatic in origin?**

This assumption is based on the young ages of the samples not allowing for significant build-up of radiogenic and nucleogenic He.

As we assumed the He$^4$ we used a $^3$He/$^4$He ratio of 5 Ra (Ra = 1.384±10-6 being the atmospheric ratio) to correct the measured $^3$He concentrations (Line 242 of the original manuscript and Blard et al (2013) *EPSL* for more information).

**Lines 284-294. The offset moraines and channels mentioned are not visually apparent in Figure 5 or verifiable using independent satellite imagery (like Google Earth Pro). Please provide more detailed information on the methodology used to project the moraine ridges onto Figure 5c so that these features can be verified.**

We believe that we show a good example of an offset moraine in Figure 5c. The projections were estimated using a best fit to the break in slope at the moraine crest as indicated by contour lines extracted from the Pleiades DTM. We have edited the methods section to explain this in more detail. The offsets are not clearly visible in Google Earth due to its lower resolution, which is why we obtained the Pleiades dataset. We have now made the Pleiades DTM and our selected moraine projections (shapefile) available in the supplemental data repository associated with this manuscript so it is available for an independent interpretation (see Data Availability section for link).

**Figure 5. Can you color the slope maps in panels C and D?**

Thank you for the suggestion. We tried changing the slope map to a colour scale, but this change makes it more difficult to see the other layers of the map.

**Figure 7. This is a really nice Figure!**

Thank you!

**Line 375 Grammar "...of the of the fault exposures..."**

Good catch, we have made the suggested edit.

**Section 5.4 After carefully inspecting Google Earth, one can notice that the area of the Polylepis Fault is made by more than one structure that would be worth including in your Figure 8. The new imagery provides a better contrast to aim in the identification of the fault traces.**

Thank you for pointing this out, this is a great observation. We have now included this new fault segment in our Figure 8 and describe it in Section 5.4. We have kept the old imagery as it shows the main structure.

**Line 399. Remove "...more..." to avoid redundancy**

Removed "more".

**Line 400. Why are these structures considered fault systems? Also, uou mention three fault systems but only two faults: Polylepis and Reservoir.**

We have edited the text to say "faults" instead of fault systems. We now explicitly mention the July 25 fault as well.

**Line 403. Shouldn't the two faults be one fault system if that's the case?**

See above response.

**Line 404. How did you define the width of the fault zone?**

We are just defining the width of the three parallel faults here. The perpendicular distance between the two furthest apart faults.

**Line 405. What do you mean by right-lateral strain?**

We are referring to the 3 mm/yr of strain modeled in the geodetic block model. We have edited this sentence to make this clearer.

**Lines 406-407. As mentioned above, please consider my comments on the offset features and the uncertainty around the interpretation of recent displacements along the Reservoir Fault.**

We believe by placing a very large uncertainty on our offset measurements, and using a wide range of ages for lateral moraines, we are adequately constraining the uncertainty in our slip rate estimations.

**Line 416. This is the first time you mention the July 25 fault in the main body of the manuscript. Please ensure the fault is named and described consistently upon its first relevant mention. Furthermore, I suggest replacing this name with a geographically significant name to facilitate referencing.**

Thank you for your suggestions. The fault in this case was delineated using the InSAR surface rupture. We agree that this was not clear in this section previously and we now explicitly mention its name here. We have kept the name "July 25 fault".

**Line 418. You didn't calculate strain rate but slip rates. Please review the article to avoid mixing concepts such as slip rate and strain rate, which are not the same.**

Thank you for pointing this out. We have changed "geologic strain rate" to "geologic slip rate"

**Lines 420-423. An examination of the mapped structures and your interpretation (Fig. 2) suggests a plausible southern extension of the Romeral Shear Zone (Buesaco and Aranda faults). Specifically, lineaments southwest of the Galeras volcano (potentially belonging to the Cauca-Patía system?) appear to show a southward prolongation, possibly merging with the Reservoir and Polylepis Faults. Given that both the Cauca-Patía and Romeral structures are right-lateral faults, their regional kinematics align well with the mapped structures in the study area. If this is true, it poses a significant tectonic implications: this extended zone may delineate either a separate, unmapped microblock or represent the northern boundary/extension of the QL microblock.**

Great point. We do consider that these structures may be the northern boundary of the microblock in Figure 9 of the discussion. We have added the fact that they are likely reactivating basement structures in a new paragraph in the Discussion section.

**Lines 428-430. This interpretation is out of place because it hasn't been developed before. Only in section 6.2 you discuss the potential influence of the CCN-VC.**

We have added "see section below for further discussion" to point the reader to where they can be introduced in detail to our interpretation. We also cite two papers as examples.

**Lines 437-443. The role of volcanic inflation should be more elaborated since it's a key conditioning factor for the triggering of the July earthquake. It'd be helpful to see that on a map or a new schematic figure.**

See reply to comment on volcanic inflation below.

**Lines 441-442. I'd recommend to remove this interpretation about the 1868 earthquake since its speculative.**

We have edited the sentence in question to say "We **speculate** that the larger M 6.6 1868 earthquake also been related to volcanism at Chiles-Cerro Negro as volcanic activity was reported around that time".

**Line 456-458. The presence of thick volcaniclastic deposits inherently presents significant difficulty in observing recent faulting. Consequently, the lack of clear surface expression through the sedimentary cover does not preclude the underlying structure from being active. For example, you didn't observe a rupture for the July 25 earthquake. So the same reasoning should be valid for the N-NE suture zones.**

We agree that there are probably active structures that we cannot observe given the thick volcanic deposits. However, we can only discuss the results that we observe, and we did not make observations of major reactivated N-NE suture zones in our study.

**Line 458. Please refer to my recommendation and include in your analysis the Romeral and Cauca-Patia fault systems which in southern Colombia are considered tectonically active. The Buesaco and Aranda faults are part of the Romeral fault system (or shear zone depending on the authors).**
See response above to comments on the Romeral fault system.

**Lines 462-465. Please rewrite this section as the current discussion is unclear. Furthermore, to support the proposed earthquake conditioning factors, please add a figure showing the modeled horizontal stress field from the CCN-Vc and its orientation relative to your mapped fault structures.**
We have modified our interpretations and rewritten the section on how volcanic activity may affect earthquakes on the studied fault. We still interpret that eastward directed motion of the crust away from the Chiles-Cerro Negro volcano would induce northwest--southeast shortening near the July 25 fault, thus promote right-lateral earthquake. However, given the different locations and orientations of the Reservoir and Polylepis fault, we suggest that inflation or deflation may play a different role in earthquake triggering, potentially through an increase in pore fluid pressure (e.g. Ebmeier et al., 2014).

**Line 470 Avoid contractions (don't)**
Edited.

**Lines 488-494. I'd recommend to keep only the radiocarbon-based paleoearthquake estimates for the Reservoir Fault. The moraine and cosmogenic 3He estimates are too vague; their 2σ uncertainty (2 kyr) is too large to constrain the recurrence interval defined by the radiocarbon dating.**
We only use radiocarbon dates to define earthquake ages and recurrence intervals in this study. We use the cosmogenic ages to confirm that the mapped moraines were most likely from the LGM and we then use them combined with the published range of ages for the LGM in this region in our slip rate calculations. We believe that this appropriately constrains the uncertainty in moraine age for these calculations.

We thank the reviewer for their comments and suggestions, we believe they have greatly strengthened the manuscript.

Nicolas Harrichhausen et al.